# PhysPDE: Rethinking PDE Discovery and a Physical Hypothesis Selection Benchmark

**Mingquan Feng[1], Yixin Huang[1], Yizhou Liu[1], Bofang Jiang[2], Junchi Yan[1]\***
[1]Sch. of Computer Science & Sch. of Artificial Intelligence, Shanghai Jiao Tong University
[2]Sch. of Physics and Astronomy, Shanghai Jiao Tong University
{fengmingquan, cindyh1103, jiangbofang, yanjunchi}@sjtu.edu.cn
https://github.com/FengMingquan-sjtu/PhysPDE

## Abstract

Existing works on recovering PDE expressions from experimental observations often involve symbolic regression. This method generally lacks the explicit incorporation of physical insights, which weaken the interpretations and effectiveness, especially in the presence of large noises. Recognizing that the primary interest of Machine Learning for Science (ML4Sci) often lies in understanding the underlying physical mechanisms or even discovering new physical laws rather than simply obtaining mathematical expressions, this paper introduces a novel ML4Sci task paradigm. It focuses on interpreting experimental data within the framework of prior physical hypotheses and theories, thereby guiding and constraining the discovery of PDE expressions. Technically, the approach is formulated as a nonlinear mixed-integer programming (MIP) problem, addressed through an efficient search scheme developed for this purpose. The experimental results on our newly designed Fluid Mechanics and Laser Fusion datasets demonstrate the interpretability and feasibility of our method. Source code and benchmarks are publicly available.

## 1 Introduction

Partial Differential Equations (PDEs) serve as powerful and pervasive tools for comprehending and describing intricate systems across physics, engineering, applied mathematics, and various other domains. Instead of solving PDEs, there has been a growing interest in the task of recovering unknown mathematical equations from observed data, especially within the machine learning community. This endeavor is particularly valuable when the observed data is noisy and imperfect, mirroring real-world conditions. However, existing techniques ranging from reinforcement learning (Du et al., 2023), genetic programming (Chen et al., 2022), and sparse regression (Wentz & Doostan, 2023) exhibit a heightened sensitivity to data noise. It is crucial to note that previous research has simply focused on producing a combination of mathematical symbols and operations while disregarding their fundamental physical implications. In practice, users of PDE recovery tools typically seek to grasp the fundamental physical mechanism rather than merely fitting mathematical models, let alone their fragility to noise. Additionally, note that the field has yet to uncover any new physical laws or PDE systems through the existing literature on PDE discovery. Based on our understanding, physicists typically do not require 're-discovering' expressions, as they already possess well-established expressions in most scenarios. Furthermore, even when they identify new expressions through purely data-driven methods, interpreting these findings within a physics framework remains essential. Ultimately, their fundamental pursuit lies in the physical interpretation of observed data.

To bridge the gap between mathematical expressions and physical interpretability for data-driven PDE recovery, we propose an innovative strategy. It leverages physical hypotheses and laws to interpret observational data to uncover new expressions that enhance our understanding of observed systems. These task paradigms and algorithms are compared in Figure 1. Take fluid dynamics for example, our PhysPDE paradigm is based on the conservation of mass, momentum, and energy, and candidate hypotheses such as constant viscosity (Newtonian fluid). The output of our paradigm is PDEs along

---

*Correspondence author who is also affiliated with Shanghai Innovation Institute. This work was partly supported by NSFC (62222607, 623B1009).

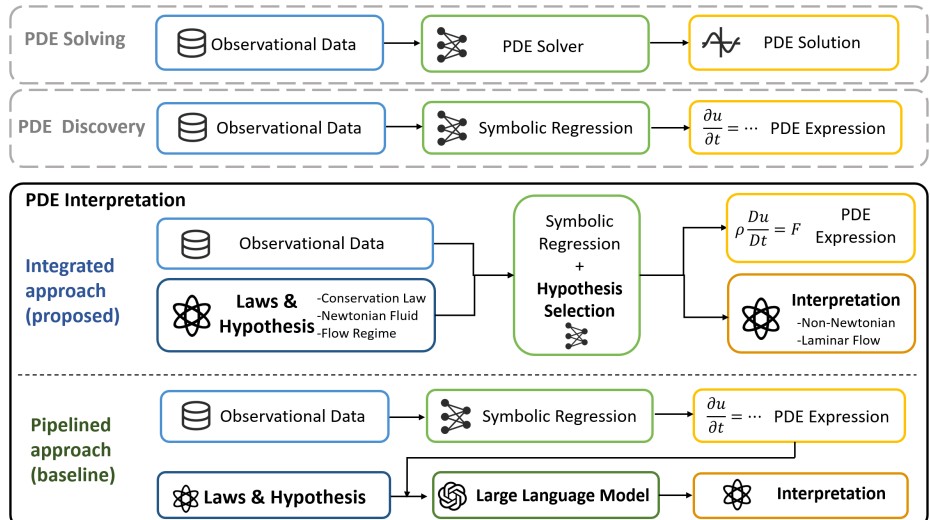

Figure 1: Comparison of different PDE learning paradigms. This paper focuses on the PDE Interpretation setting in the bottom box, which, to our best knowledge, is in contrast to existing literature devoted to the other two settings. Although a pipeline of existing algorithms can interpret PDE data to some extent, we propose a stronger integrated approach directly solving the combinatorial problem.

with selected hypotheses. This paradigm shift seeks to align mathematical models more closely with their underlying physical realities, facilitating a deeper comprehension of complex phenomena.

The possible applications include inferring key factors affecting the efficiency of heat exchangers (e.g., turbulence) (Wang et al., 2023) and aiding in the diagnosis of physical phenomena in nuclear fusion that cannot be directly measured (e.g., magnetic fields) (Peebles et al., 2022). In these applications, existing mathematical formulas and hypotheses are relatively comprehensive and do not require "rediscovery." What scientists need is to use this knowledge to explain experimental results. **The highlights of this paper include:**

**1) New Task Formulation:** Going beyond the prevalent regression of PDE expressions from observations in existing literature, in this paper we introduce a new ML4Sci task, which *standing upon the shoulders of* a suite of basic physical laws and hypotheses, employing learning to identify the hypothesis that best aligns with the data.

**2) New Algorithms:** Besides a simple pipeline of existing models, we design a new integrated algorithm for the task. We formulate the task above as a bi-level mixed-integer optimization problem, consisting of two layers: hypothesis selection at the outer level and expression recovery at the inner.

**3) New Datasets and Experiments Design:** Following the proposed paradigm, we present two new datasets, Fluid Mechanics and Laser Fusion. Experiments across multiple scenarios demonstrate the feasibility and effectiveness of our approach.

## 2   RELATED WORKS

**PDE Solving.** Solving PDEs that describe complex system dynamics is crucial in scientific disciplines like computational physics. The goal is to find a solution function that satisfies the equations given initial and boundary conditions. Representative datasets include PDEBench (Takamoto et al., 2022), PDEArena (Gupta & Brandstetter, 2022) , and CFDBench (Luo et al., 2023). Classic methodologies, including finite difference (Strikwerda, 2004), finite volume (Eymard et al., 2000), and finite element (Zienkiewicz et al., 2005), concentrate on devising numerical schemes that simplify the problem into linear equation systems. Deep-learning-based solvers such as PINN (Raissi et al., 2019) and DeepBSDE (Han et al., 2018) have been emerging, extending the capability of numerical solvers.

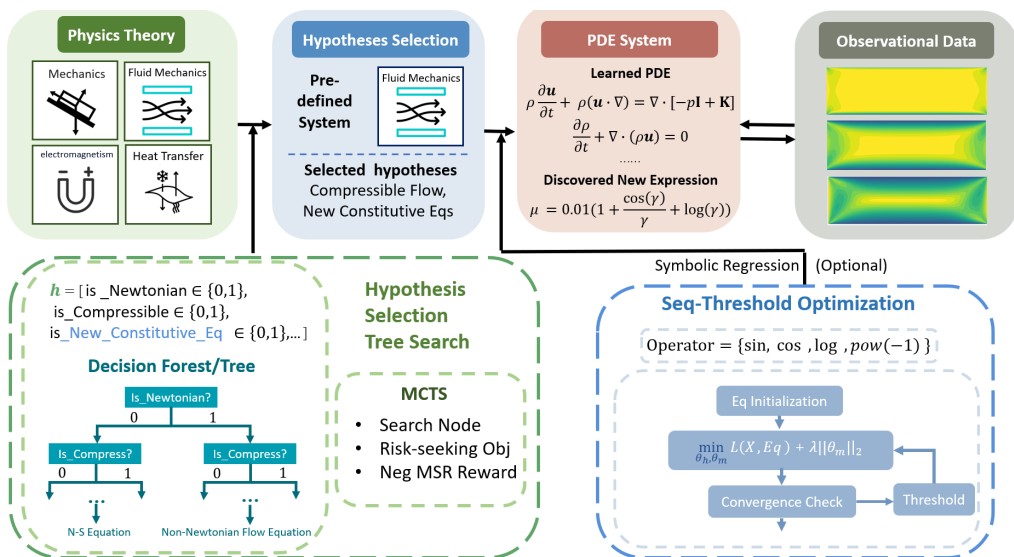

Figure 2: Overview of the proposed hypothesis selection framework, selecting hypothesis and constructing PDE system from both physics theory and observational data. The constructed PDE system consists of equations from selected physics hypotheses and newly discovered expressions.

Table 1: Comparisons with related PDE solving, discovery, and interpretation tasks in physics (Takamoto et al., 2022; Raissi et al., 2019; Bhamidipaty et al., 2024; Rudy et al., 2017). To our knowledge, our work provides the first dataset and algorithm for the PDE interpretation task.

| Task | Physics Sub-discipline | Dataset/Tool | Algorithm |
|---|---|---|---|
| PDE Solving | Computational Physics | PDEBench | PINN |
| PDE Discovery | Phenomenology | DynaDojo | PDE-FIND |
| **PDE Interpretation** (ours) | Theoretical Physics | **PhysPDE** | **HSTS** |

**PDE Discovery.** In the domain of physics phenomenology, the pivotal problem is deducing the mathematical symbolic expressions (i.e. symbolic regression) of PDE systems from observed data. There is currently no specifically designed PDE Discovery Benchmark. However, DynaDojo (Bhamidipaty et al., 2024), as a general-purpose PDE Benchmark, can be used to some extent for this purpose. Data-driven methodologies, including reinforcement learning (Du et al., 2023), genetic programming (Xu et al., 2020; Chen et al., 2022), and sparse regression (Rudy et al., 2017; Messenger & Bortz, 2021; Wentz & Doostan, 2023), have been prominently applied to this end. These approaches adeptly identify the mathematical constructs of PDEs but frequently overlook the integration of physical principles.

**PDE Interpretation.** A critical examination of existing data-driven methodologies for PDE discovery underscores a pervasive limitation: their insufficient focus on ensuring interpretability within a concrete scientific theory framework (e.g. physical theorems). This oversight, as highlighted by Zhang et al. (2023) and Faroughi et al. (2023), manifests a pronounced divide between mathematical precision and physical interpretability. Further exploration by Cornelio et al. (2023) and Cory-Wright et al. (2023) accentuates this concern, advocating for a deeper comprehension of the physical underpinnings of equations beyond their mere mathematical representation. Despite these insights, to the best of our knowledge, a universally recognized methodology for the systematic interpretation of nonlinear PDE system data remains elusive.

## 3 PDE INTERPRETATION TASK FORMULATION

This study introduces PDE interpretation, a novel task aimed at the recovery of physics hypotheses as well as expressions from observational data. The cornerstone named Hypothesis Selection, distin-

guished by its capacity to conceptualize systems of PDEs as amalgamations of physics hypotheses and theories, effectively narrows the chasm between physical understanding and mathematical depiction. The subsequent section delineates the technical aspects of our problem formulation and method.

Our approach is grounded in two fundamental physics principles: (i) physics formulas, like PDEs/ODEs, stem from physics laws, and (ii) physics laws are derived from hypotheses and basic theories (Carcassi & Aidala, 2022). We aim to recover *valid* PDEs that adhere to underlying hypotheses, moving beyond mere data fitting. We define our task as selecting the best-fit set of hypotheses from a "hypothesis universe" to fit a system of PDEs, conceptualized as a two-level min-min optimization:

$$\min_{\boldsymbol{h}, \boldsymbol{\theta_h}} \min_{\boldsymbol{m}, \boldsymbol{\theta_m}} \quad \ell(X, Eq_{\boldsymbol{h}, \boldsymbol{m}, \boldsymbol{\theta_h}, \boldsymbol{\theta_m}})$$
$$\text{s.t.} \quad C(\boldsymbol{h}, \boldsymbol{m}, \boldsymbol{\theta_h}, \boldsymbol{\theta_m}) \geq 0 \tag{1}$$

**Outer level (hypothesis selection)**: The outer optimization targets the decision variable $\boldsymbol{h}$ and physics parameter $\boldsymbol{\theta_h}$. $\boldsymbol{h}$ is a binary vector encoding the selection of physics hypotheses, while $\boldsymbol{\theta_h}$ is a vector comprising undetermined coefficients and constants of the system.

**Inner level (symbolic regression, optional)**: Selecting the "discover new partial expressions" hypothesis triggers the inner optimization level, aimed at discovering new expressions within a PDE expression segment. The algorithm can modify the equation directly but is confined only to the specific part of the equation that the selected hypothesis allows. At this level, $\boldsymbol{m}$, being an integer vector, denotes selections from a library of basis functions or operations. These choices instantiate a mathematical expression, characterized by coefficients represented by the real vector $\boldsymbol{\theta_m}$.

The combined decisions $\boldsymbol{h}, \boldsymbol{m}$ and parameters $\boldsymbol{\theta_h}, \boldsymbol{\theta_m}$, informed by physics prior knowledge, collectively define the PDE system $Eq$ and its constraints $C$. The abstract loss function is:

$$\ell(X, Eq) = \text{MSR}(X, Eq) + \text{Reg}(Eq). \tag{2}$$

We combine the mean squared residual (MSR) between observation and recovered system and regularization (Reg) to enforce simplicity per Occam's razor principle (Walsh, 1979). For brevity, the arguments of both $Eq$ and $C$ are omitted. The constraints $C$ encode the physical validity constraints of variables, for example, the mutual exclusiveness of hypotheses, and bounds of parameters. This problem framework is versatile, accommodating various solvers. We next provide a detailed account of our specific definition and algorithmic approach.

**Decision Forest.** The hypotheses exhibit complex inter-dependencies, e.g. mutual exclusiveness and hierarchy. We opt to represent these dependencies via a decision forest rather than as constraints in the optimization problem Eq. 1, aiming to improve readability and streamline the search.

We define a decision as a set of choices among a group of mutually exclusive hypotheses. For instance, the decision labeled `Is_Newtonian` encompasses two hypotheses, `Newtonian_Fluid` and `Non_Newtonian Fluid`. Subsequent decisions are defined as those influenced by the choices made in the preceding decisions. For instance, the decision `Type_Non_Newtonian` is induced by the second choice made in the `Is_Newtonian` decision. Once the non-Newtonian hypothesis is selected, all descendant searches are non-Newtonian. Based on the above definitions, we organize decisions hierarchically as a forest, with decisions being nodes. Subsequent decisions are marked as child nodes of the preceding decisions. Nodes lacking parent nodes are designated as root nodes.

Under this framework, a closed subset of physics hypotheses constitutes a tree, while the aggregate of these subsets forms a forest. A root-to-leaf path in any tree represents a partial decision instance, and the ensemble of such paths across all trees defines a complete decision instance. Given the forest, there is a one-to-one mapping between the full decision instance and a valid hypothesis vector $\boldsymbol{h}$.

A complete decision, combined with existing physics theorems, can uniquely derive a system of PDEs. We automatically implement such derivation via the symbolic computing tool (Meurer et al., 2017). This encoding scheme allows for a comprehensive representation of physical concepts and their mathematical counterparts. Figure 2 illustrates an example, showcasing the decomposition of compressible Navier-Stokes equations into two hypotheses: **Compressibility** and **Newtonian fluid**. Figure 3 visualizes the details of PDE construction from hypotheses.

The decision forest is a simplified model postulating that each decision node has a single parent and that hypotheses across decisions are mutually inclusive. In fluid mechanics as considered in the paper

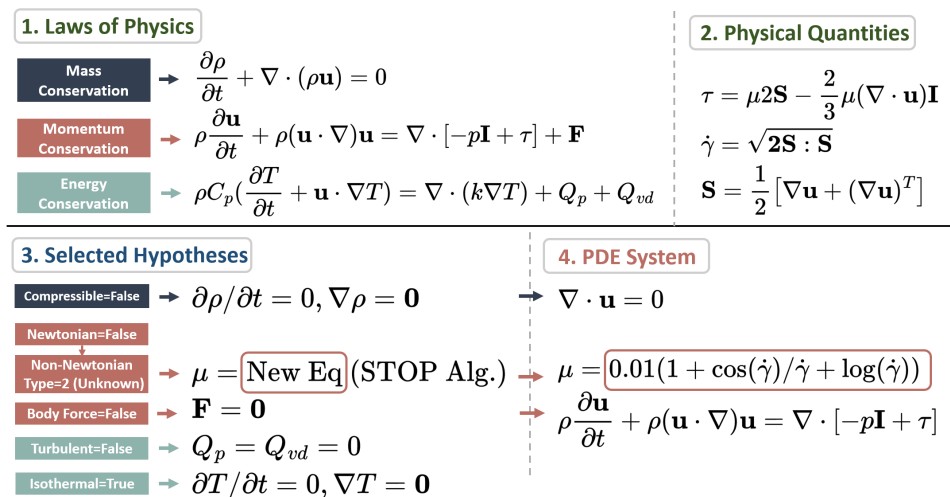

Figure 3: Given observational data and physics theories including 1) laws of physics, 2) physical quantities, and a search space of hypotheses forest (not shown here), our algorithm can provide a best-fit set of 3) selected hypotheses and 4) PDE system, possibly with newly discovered expressions. We start from the full PDE system of the unchangeable physics theorem (e.g. conservation laws in fluid mechanics). On top of that, the derivation is done by plugging the expression of the selected hypothesis into the PDE of conservation laws. The example PDE is an incompressible version of our S2 Fluid Mechanics dataset. A brief introduction to fluid mechanics and the construction of these equations is provided in Appendix C.

for example, exceptions to these hypotheses are addressed by an ad-hoc filter function for simplicity. Modeling these exceptions accurately may require a more sophisticated structure, e.g. a decision graph, which we leave for future work.

**Loss Function.** The loss function $\ell$ is defined with:

$$\text{MSR}(X, Eq) = \sum_{i,j} \frac{[\text{Pool}(Eq^{j,L}(X_i) - Eq^{j,R}(X_i))]^2}{\text{size}(X)}, \tag{3}$$

$$\text{Reg}(Eq) = \lambda_{Eq}\,\text{size}(Eq) + \lambda_{\boldsymbol{h}}\,\text{size}(\boldsymbol{h}) + \lambda_{\boldsymbol{\theta}_{\boldsymbol{h}}}\|\boldsymbol{\theta}_{\boldsymbol{h}}\|_0 + \lambda_{\boldsymbol{m}}\|\boldsymbol{m}\|_0, \tag{4}$$

where $\lambda$ denotes hyperparameters. It calculates the residual as the averaged squared differences between the left $L$ and right $R$ sides of each equation $Eq^j$, among all collocation points $X_i$. Average pooling Pool - a simple weak-form (Messenger & Bortz, 2021), serves as a de-noising technique. The regularization term $\text{Reg}(Eq)$ incorporates the equation system's complexity of hypothesis selection and new expression discovery. The former is quantified by a weighted sum of the total equation size, decision vector size, and count of active physics parameters while the latter is measured by the sparsity of the chosen basis functions or operators.

Both the outer hypothesis selection and inner symbolic regression problems have exponentially large search spaces, necessitating specially designed baseline algorithms. We developed a Monte Carlo Tree Search (MCTS)-inspired algorithm called Hypothesis Selection Tree Search (HSTS) for the outer problem, and a Sequential Threshold OPtimization (STOP) algorithm for the inner problem. Due to space limitations, detailed designs of these algorithms are provided in Appendix F.

## 4 FLUID MECHANICS DATASET AND EXPERIMENTS

### 4.1 EXPERIMENT SETUP

**Physics Hypotheses.** In our research, while the proposed methodological framework is adaptable across multiple physics domains, we specifically target fluid mechanics, a focal area within learning-based PDE discovery studies. We design 16 decision points across 38 hypotheses based on textbook

(Batchelor, 1967), encapsulating fundamental physics concepts such as flow regimes, compressibility, constitutive equations, and non-isothermal flows. The validity of our design is also acknowledged by the mathematician expert in this area. We also explore hypotheses concerning new expressions of constitutive equations. Details on the complete set of hypotheses are provided in Appendix A.

**Scenarios.** We design three different scenarios constructed from three representative sets of hypotheses: 1) **S1**: Newtonian Fluid, 2) **S2**: Non-Newtonian Fluid with Unknown Constitutive Equation (Figure 3), 3) **S3**: Non-isothermal Flow

Scenario S1 elaborates on the derivation of the Navier-Stokes equations, which are foundational to fluid mechanics (Constantin & Foiaş, 1988). The flow is set to be compressible, isothermal, non-turbulent, and non-gravity. The viscosity is constant (Newtonian fluid assumption) $\mu = 0.01$. Scenario S2, a variation of S1, evaluates the model's capability of discovering new expressions by substituting the constitutive equation $\mu = 0.01$ in S1 with an unknown new expression $\mu = 0.01 + 0.01 \cos(\gamma)/\gamma + 0.01 \log(\gamma)$, where $\gamma$ is the shear rate. S3 deviates from S1 by switching to the non-isothermal hypothesis. It leads to thermal-hydraulic equations, which are widely applied in nuclear technology (Zhang et al., 2018) and geophysics (Bächler & Kohl, 2005). The derived PDEs for these scenarios are solved via COMSOL Multiphysics [1], a commercial simulation software. Observable physics variables include velocity, density, pressure, and temperature, each modeled as two-dimensional in both space and time. For each scenario, datasets for training and testing are generated under varying boundary conditions, ensuring identical sizes.

**Performance Metrics.** We use the recall and precision metric. The recall is defined as the percentage of the number of correctly learned decision over the true decision, $R = \|\boldsymbol{b}_{learn} \odot \boldsymbol{b}_{true}\|_0 / \|\boldsymbol{b}_{true}\|_0$, and precision is $P = \|\boldsymbol{b}_{learn} \odot \boldsymbol{b}_{true}\|_0 / \|\boldsymbol{b}_{learn}\|_0$, where $\odot$ is element-wise product. Vectors $\boldsymbol{b}$ are binary vectors denoting the selection of hypotheses or PDE terms. Accuracy is measured by MSR on dataset splits and efficiency by training wall-time.

## 4.2 Data Generation and Sampling

**S1 and S2.** The study investigates fluid dynamics within a rectangular domain of dimensions $1\,\mathrm{m} \times 0.3\,\mathrm{m}$, with the fluid initially set to a density of $1\,\mathrm{kg/m}^3$, x-velocity $\sin(\pi x)$, y-velocity $\sin\left(\frac{\pi y}{0.3}\right)$, and pressure at zero. Boundary conditions are implemented as no-slip walls, with the bottom wall moving left at 1 m/s, and the top wall moving right at 1 m/s for Scenario 1 (S1) and 2 m/s for Scenario 2 (S2), with pressure fixed at zero at the bottom-left corner. The simulation employs a finite-element method on a mesh resulting in approximately 30,000 degrees of freedom, with results sampled on a $100 \times 100$ grid over a temporal range(0, 0.004, 1).

**S3.** This scenarios is adapted from COMSOL Application Gallery [2]. The study models fluid dynamics within a $0.005\,\mathrm{m} \times 0.04\,\mathrm{m}$ rectangular domain, initially set with a density of $997\,\mathrm{kg/m}^3$, velocity of zero, pressure $\rho g(0.04 - y)$, and temperature $293.15\,\mathrm{K}$. A half-circle heater of radius $0.0025\,\mathrm{m}$, positioned at $(0, 0.015)$, heats the fluid. Boundary conditions are periodic on the left and right, an inlet at the bottom with a $0.005$ m/s inflow velocity and $293.15\,\mathrm{K}$ temperature, and an outlet at the top designed to suppress backflow. Heater temperatures are set to $303.15\,\mathrm{K}$ in Scenario 1 (S1) and $323.15\,\mathrm{K}$ in Scenario 2 (S2). The numerical solution employs a finite-element method focusing on the upper half of the rectangular domain, encompassing approximately 25,000 degrees of freedom. Results are sampled over a $50 \times 100$ spatial grid within a temporal range $(1, 0.005, 3)$.

## 4.3 Hyper-Parameters Setting

**Loss** The pooling size is 5. The coefficients are $\lambda_{Eq} = 10^{-7}$, $\lambda_{\boldsymbol{\theta_h}} = 10^{-5}$, $\lambda_{\boldsymbol{h}} = 10^{-5}$, $\lambda_{\boldsymbol{m}} = 10^{-5}$.

**HSTS** For S1 and S2, the maximum number of rollout is 30, the $\epsilon$ of risk-seeking object is 0.5; while in the S3, the the maximum number of rollout is 60 and $\epsilon = 0.05$. The bound $l_{\min} = 0.0001$, $l_{\max} = 1$. The exploration weight in the UCT is 1. The 10-core multi-processing is applied at node level, using the virtual loss technique.

**STOP** We set the maximum iteration depth at 3, the $l_2$ regularization $\lambda_{\boldsymbol{\theta_m}} = 0.01$, and the weight tolerance $T = 0.005$. The parameters $\boldsymbol{\theta_m}$ are scaled to $[0.05, 0.05]$ for numerical stability.

---

[1] www.comsol.com
[2] https://www.comsol.com/model/heat-transfer-by-free-convection-122

**PDE Parameter Estimation** The optimization algorithm is SLSQP implemented in SciPy (Virtanen et al., 2020), with a timeout of 5 minutes for S1 and S2, and 15 minutes for S3. The parameters are first initialized from normal distribution $N(0, 1)$. For parameters with different orders of magnitude, e.g. the heat capacity at constant pressure $C_p \in [4100, 4300]$, we further add biases to the initialization, e.g. a bias of 4200 in $C_p$ case. The biases and parameter ranges are informed by physics knowledge.

**Experiment Protocols.** We perform 5-fold cross-validation on the training set for performance estimation, highlighting the top three decisions with the highest average validation loss. The hyper-parameters of the search algorithm are chosen without tuning, and variable initialization $\theta$ follows a normal distribution. Computations are executed on a remote server using 10 CPU cores concurrently. For the STOP algorithm, the function library comprises second-order polynomials of $\{\sin, \cos, \log, pow(-1)\}$. To restrict the wall-time of ES, we limit its maximum wall-time to 24 hours, only include the new expression hypothesis in the search space in S2, and incorporate STOP for internal expression discovery. Other numerical settings, including hyper-parameter configurations, are detailed in Appendix 4.3.

**Baselines.** As there is no baseline directly available for our hypothesis selection problem, we designed an Exhaustive Search (ES) baseline that randomly enumerates the search space to demonstrate the efficiency of the proposed search algorithm. A variant of ES is ES-TL (Exhaustive Search with Time Limit), which terminates the ES under the same runtime as our method. Also, the performance of the Ground Truth (GT) hypothesis is reported to evaluate the accuracy of parameter estimation. For the PDE symbolic regression baseline, we adopt the popular PDE-FIND (Rudy et al., 2017), which is still in active maintenance by the open-source community. We also tried other more recent peer methods like genetics algorithm based SGA-PDE (Chen et al., 2022) and reinforcement learning based DISCOVER (Du et al., 2023) yet it remains in vain and these methods perform less competitive than PDE-FIND in our challenging setting. We further apply **GPT-4o** (Achiam et al., 2023), a Large Language Model (LLM), to extract physics properties from the underlying system. In our pre-experiment, GPT-4o alone cannot infer the physics hypothesis directly from raw data, possibly due to the large input size and combinatorial nature of the task. Thus, we fed PDE expressions (discovered by SINDy or configured manually) and physics hypothesis priors into GPT-4o to infer the correct physics hypothesis, corresponding to the pipelined approach in the Figure 1.

## 4.4 RESULTS AND DISCUSSION

We evaluate our framework by answering the following research questions (RQs):

**RQ1**: Does HSTS select the hypotheses correctly and efficiently?

**RQ2**: Is HSTS able to recover known PDEs and to discover unknown expressions?

**RQ3**: Do the PDEs recovered by HSTS have acceptable interpolation and extrapolation accuracy?

**RQ4**: How does the pipelined baseline compare with the HSTS (integrated approach) in physics interpretation?

**(RQ1) Hypotheses Selection.** Table 2,3, and 4 resent the hypothesis selection performance comparison results of HSTS with the ES baseline. With 3500 valid decision combinations in the search space, HSTS successfully identifies nearly all hypotheses across all scenarios, achieving the same top-1 recall and precision as ES, demonstrating its superior capability in terms of physics interpretability. Notably, HSTS's training wall-time is significantly lower than ES's, by an order of magnitude, highlighting the efficiency of our search scheme. In Table 2, HSTS outperforms ES in top-2 and top-3 performance metrics, which is attributed to HSTS's strategy of extending the search space locally, resulting in a clustering of visited nodes and similar top results. For instance, in scenario S1, HSTS's top-1 result is the correct solution, with its top-2 and top-3 variations differing by only one or two hypotheses. In contrast, ES's global search approach tends to yield more diverse top results. Detailed listings of the selected hypotheses for each scenario are provided in Appendix D.

**(RQ2) PDE Recovery and Discovery.** Table 9 showcases the recovered PDE terms and parameters, while Table 5 compares the performance of the HSTS (top-1) with that of PDE-FIND in identifying PDE terms. Our results demonstrate that HSTS accurately recovers the majority of terms and parameters across all scenarios, including successfully identifying new expressions in S2. Conversely, PDE-FIND tends to overfit the data with overly complex expressions, resulting in low recall and

Table 2: RQ1 on S1: Hypothesis Selection.

| Model | | ↑ Recall | ↑ Precision | ↓ Wall-time(h) |
|---|---|---|---|---|
| Ours | Top-1 | **1.00** | **1.00** | |
| | Top-2 | **0.86** | **0.86** | **0.14** |
| | Top-3 | 0.71 | **0.71** | |
| ES-TL | Top-1 | 0.86 | 0.55 | |
| | Top-2 | 0.71 | 0.45 | 0.28 |
| | Top-3 | 0.57 | 0.31 | |
| ES | Top-1 | 1.00 | 1.00 | |
| | Top-2 | 0.86 | 0.55 | 16.68 |
| | Top-3 | **0.86** | 0.55 | |

Table 3: RQ1 on S2: Hypothesis Selection.

| Model | | ↑ Recall | ↑ Precision | ↓ Wall-time(h) |
|---|---|---|---|---|
| Ours | Top-1 | **1.00** | **1.00** | |
| | Top-2 | **0.86** | **0.86** | **4.10** |
| | Top-3 | **0.86** | **0.55** | |
| ES-TL+ STOP | Top-1 | 0.71 | 0.56 | |
| | Top-2 | 0.57 | 0.31 | 4.36 |
| | Top-3 | 0.57 | 0.31 | |
| ES+ STOP | Top-1 | 1.00 | 1.00 | |
| | Top-2 | 0.86 | 0.55 | 24.00 (Timeout) |
| | Top-3 | 0.86 | 0.55 | |

Table 4: RQ1 on S3: Hypothesis Selection.

| Model | | ↑ Recall | ↑ Precision | ↓ Wall-time(h) |
|---|---|---|---|---|
| Ours | Top-1 | **0.91** | **0.91** | |
| | Top-2 | **1.00** | **1.00** | **3.12** |
| | Top-3 | **0.82** | **0.82** | |
| ES-TL | Top-1 | 0.82 | 0.82 | |
| | Top-2 | 0.73 | 0.62 | 3.30 |
| | Top-3 | 0.82 | 0.82 | |
| ES | Top-1 | 0.91 | 0.91 | |
| | Top-2 | 1.00 | 1.00 | 21.11 |
| | Top-3 | 0.82 | 0.82 | |

Table 5: RQ2 on all Scenarios.

| Model | Scenario | ↑ Recall | ↑ Precision | ↓ Complexity |
|---|---|---|---|---|
| Ours | S1 | **1.00** | **1.00** | **109** |
| | S2 | **1.00** | **1.00** | **360** |
| | S3 | **1.00** | **0.93** | **180** |
| PDE-FIND | S1 | 0.88 | 0.04 | 1423 |
| | S2 | 0.20 | 0.02 | 1377 |
| | S3 | 0.22 | 0.01 | 2176 |

Table 6: RQ3 on S1: Mean Squared Residuals.

| Model | Train | Valid | Test |
|---|---|---|---|
| Ours | $2.80 \times 10^{-4}$ | $5.22 \times 10^{-4}$ | $4.83 \times 10^{-4}$ |
| GT | $2.89 \times 10^{-4}$ | $5.09 \times 10^{-4}$ | $\mathbf{4.50 \times 10^{-4}}$ |
| PDE-FIND | $\mathbf{1.58 \times 10^{-4}}$ | $\mathbf{2.79 \times 10^{-4}}$ | $1.48 \times 10^{-1}$ |
| DISCOVER | 0.123 | 0.132 | 1.231 |
| SGA-PDE | 24.43 | - | - |

Table 7: RQ3 on S2: Mean Squared Residuals.

| Model | Train | Valid | Test |
|---|---|---|---|
| Ours | $\mathbf{3.24 \times 10^{-3}}$ | $4.49 \times 10^{-3}$ | $4.75 \times 10^{-3}$ |
| GT | $3.35 \times 10^{-3}$ | $\mathbf{4.38 \times 10^{-3}}$ | $\mathbf{4.29 \times 10^{-3}}$ |
| PDE-FIND | $7.21 \times 10^{-3}$ | $8.44 \times 10^{-3}$ | $1.66 \times 10^{-1}$ |
| DISCOVER | 0.144 | 0.158 | 1.232 |
| SGA-PDE | 28.7 | - | - |

Table 8: RQ3 on S3: Mean Squared Residuals.

| Model | Train | Valid | Test |
|---|---|---|---|
| Ours | $\mathbf{9.86 \times 10^{-3}}$ | $\mathbf{9.38 \times 10^{-3}}$ | $\mathbf{1.74 \times 10^{-2}}$ |
| GT | $1.03 \times 10^{-2}$ | $9.83 \times 10^{-3}$ | $1.76 \times 10^{-2}$ |
| PDE-FIND | $1.93 \times 10^{-4}$ | $2.06 \times 10^{-4}$ | $9.55 \times 10^{-1}$ |
| DISCOVER | 1.675 | 1.682 | 2.312 |

extremely low precision. Despite careful tuning of PDE-FIND's parameters and the application of denoising techniques (Messenger & Bortz, 2021), performance improvement remains elusive. We attribute PDE-FIND's limitations to its lack of embedded physics knowledge, leading to a significantly larger search space compared to HSTS and, consequently, heightened difficulty in problem-solving.

**(RQ3) Interpolation and Extrapolation.** To assess the generalization ability of our selected hypotheses and the induced PDEs, we conduct interpolation and extrapolation tests on both training and test datasets, which are generated from the same governing equations with different boundary conditions. Table 6, 7, and 8 show the MSR for all data splits. Our findings affirm that HSTS-recovered PDEs consistently show high accuracy in both interpolation and extrapolation, closely aligning with the ground truth PDEs. In contrast, PDE-FIND-recovered PDEs excel in interpolation but exhibit significantly diminished extrapolation capabilities, characterized by orders of magnitude higher MSR. Furthermore, it's noteworthy that even the true PDEs themselves exhibit non-negligible residuals, likely stemming from systematic errors in data generation, sampling, and finite difference derivative approximations. This observation, coupled with the promising results of HSTS in RQ1 and RQ2, underscores the robustness of HSTS in the presence of data noise. This property makes the algorithm well-suited for real-world scenarios where data imperfections are prevalent.

**(RQ4) Integrated vs Pipelined.** Table 10 compares the results of the physics hypothesis selection of both integrated and pipelined models. Our integrated HSTS algorithm can get almost everything correct, while the SINDy+GPT pipeline isn't able to find physics properties effectively. However, if ground truth PDE expressions are provided, GPT can infer physics better. We conjecture that the

Table 9: Examples of identified PDEs and Estimated Parameters (notations defined in Appendix B).

| | PDE-FIND | Ours | GT |
|---|---|---|---|
| S1 | $\frac{\partial u_x}{\partial t} = -160393 \frac{\partial^2 p}{\partial x \partial y} + 6878 u_y \frac{\partial p}{\partial x} \cdots$ $\frac{\partial u_y}{\partial t} = 12777 \frac{\frac{\partial p}{\partial x}}{u_y} + 11125 \frac{\frac{\partial p}{\partial y}}{u_y} \cdots$ $\frac{\partial \rho}{\partial t} = -16339 u_y \frac{\partial p}{\partial x} - 18572 \frac{1}{p} \cdots$ | $\rho \frac{\mathrm{D}\mathbf{u}}{\mathrm{D}t} = \nabla \cdot (-p\mathbf{I} + \boldsymbol{\tau})$ $\frac{\partial \rho}{\partial t} + \nabla \cdot (\rho \mathbf{u}) = 0$ $\mu = 0.01000$ | $\rho \frac{\mathrm{D}\mathbf{u}}{\mathrm{D}t} = \nabla \cdot (-p\mathbf{I} + \boldsymbol{\tau})$ $\frac{\partial \rho}{\partial t} + \nabla \cdot (\rho \mathbf{u}) = 0$ $\mu = 0.01$ |
| S2 | $\frac{\partial u_x}{\partial t} = -15580 \frac{\partial^2 p}{\partial x \partial y} + 60459 u_y \frac{\frac{\partial p}{\partial x}}{u_y} \cdots$ $\frac{\partial u_y}{\partial t} = -130897 \frac{\frac{\partial p}{\partial y}}{u_y} + 48085 \gamma \frac{\partial^2 p}{\partial y^2} \cdots$ $\frac{\partial \rho}{\partial t} = 220459 \gamma \frac{\partial p}{\partial x} + 132342 u_y \frac{\partial p}{\partial y} \cdots$ | $\rho \frac{\mathrm{D}\mathbf{u}}{\mathrm{D}t} = \nabla \cdot (-p\mathbf{I} + \boldsymbol{\tau})$ $\frac{\partial \rho}{\partial t} + \nabla \cdot (\rho \mathbf{u}) = 0$ $\mu = 0.01004 + 0.01003 \frac{\cos(\gamma)}{\gamma}$ $+0.01007 \log(\gamma))$ | $\rho \frac{\mathrm{D}\mathbf{u}}{\mathrm{D}t} = \nabla \cdot (-p\mathbf{I} + \boldsymbol{\tau})$ $\frac{\partial \rho}{\partial t} + \nabla \cdot (\rho \mathbf{u}) = 0$ $\mu = 0.01(1 + \frac{\cos(\gamma)}{\gamma} + \log(\gamma))$ |
| S3 | $\frac{\partial u_x}{\partial t} = -0.001 u_x - 0.003 u_x \frac{\partial^2 u_x}{\partial x \partial y} \cdots$ $\frac{\partial u_y}{\partial t} = -0.75 u_x p \frac{\partial u_x}{\partial y} - 0.5 p \frac{\partial^2 u_x}{\partial x^2} \cdots$ $\frac{\partial \rho}{\partial t} = -1.6 u_y \frac{\partial p}{\partial x} - 1.2 \frac{\frac{\partial^2 p}{\partial y^2}}{u_y} \cdots$ $\frac{\partial T}{\partial t} = 26.7 u_x u_y \frac{\partial^2 u_x}{\partial x \partial y} - 17.1 T \frac{\partial T}{\partial y} \cdots$ | $\rho \frac{\mathrm{D}\mathbf{u}}{\mathrm{D}t} = \nabla \cdot (-p\mathbf{I} + \boldsymbol{\tau}) + \mathbf{F}$ $\frac{\partial \rho}{\partial t} + \nabla \cdot (\rho \mathbf{u}) = 0$ $\rho C_p \frac{\mathrm{D}T}{\mathrm{D}t} = \nabla \cdot (k \nabla T)$ $+ Q_p + Q_{vd}$ $\mu = 0.00992, \ g = -9.8000$ $k = 0.58704, \ C_p = 4200.3$ $\beta = 0.15747$ | $\rho \frac{\mathrm{D}\mathbf{u}}{\mathrm{D}t} = \nabla \cdot (-p\mathbf{I} + \boldsymbol{\tau}) + \mathbf{F}$ $\frac{\partial \rho}{\partial t} + \nabla \cdot (\rho \mathbf{u}) = 0$ $\rho C_p \frac{\mathrm{D}T}{\mathrm{D}t} = \nabla \cdot (k \nabla T) + Q_{vd}$ $\mu = 0.01, \ g = -9.8$ $k = 0.6, \ C_p = 4186$ |

pipelined approach is inferior due to the limitation of information propagation. Specifically, the PDE discovery step is inaccurate due to the lack of hypothesis selection priors or feedback, and in turn, such inaccuracy bottlenecks the performance of the latter hypothesis selection step.

More details about physics interpretation and the meaning of each decision are shown in appendix D.

## 5 LASER FUSION DATASET AND EXPERIMENTS

To demonstrate the application of PhysPDE in real physics research, we experimented in the field of Fast Ignition (FI) of inertial confinement fusion (ICF) (Tabak et al., 1994), guided and assisted by experts in the domain. In the FI experiments, high-density deuterium-tritium plasma is heated by relativistic electron beams (REB) over a time scale of approximately 10 ps, raising the temperature by about 10 keV (approximately 120 million Kelvin). During this process, some physical quantities (such as magnetic fields, REB divergence angles, plasma dopant types and plasma degeneracies, etc.) are crucial for evaluating the effectiveness of the FI experiments and optimizing the design of future experiments. However, due to the extremely rapid evolution of the physical system and the difficulty of diagnosing certain physical processes occurring in high-density plasma through optical methods, these physical quantities are typically challenging to measure directly and precisely (Abu-Shawareb et al., 2022; Peebles et al., 2022). Therefore, we aim to indirectly derive these physical quantities through the evolution of directly observable quantities, such as plasma temperature. Researchers can provide prior knowledge in the form of theoretical analyses (hypotheses) about how these physical quantities influence the system's governing equations (PDEs). In this context, PhysPDE aids researchers in automating and efficiently conducting diagnostic analyses.

Leveraging domain knowledge, we developed a hypothesis decision forest (Table.15), comprising 5 decisions and 32 valid combinations. The detailed definitions and results are provided in Appendix H. In this work, we first use a numerical simulation program to simulate the FI experiment and its diagnostic process to construct a dataset. In the future, the algorithm developed in this work is expected to be applied to the ongoing double-cone collision ignition FI experiment (Zhang et al., 2020). The data were simulated using the specialized solver HEETS (Xu et al., 2019), with $80 \times 80 \times 80$ spatial grids and 26 temporal grids. HEETS is a particle-fluid hybrid simulation program specifically developed for simulating charged particle transport, nuclear reactions, and the evolution of fields in high-density plasma in FI. In this program, the motion of the REB is described using relativistic stochastic partial differential equations (Robinson et al., 2015), while background electron-ion plasmas are described by a reduced two-fluids model (Gardiner et al., 1985). We constructed the dataset by performing FI simulations over a physical time of 10 ps under various conditions, including different applied magnetic fields, REB divergence angles, target doping, and initial plasma degeneracy. Given the problem's complexity and dimensionality, each simulation required approximately 0.72h

Table 10: Results on the physics property extracted from data and expressions. Number marked red means a wrong decision is made on this property. Term colored red represents a non-existing property in the original system. Abbreviations: GT+GPT (ground truth equations are fed to ChatGPT-4o)

| | S1 | S2 | S3 |
|---|---|---|---|
| Ours | is_compressible: 1
is_isothermal: 1
is_newtonian: 0
is_turbulent: 0
type_body_force: 0
type_non_newtonian: 3
type_non_turbulent: 0 | is_compressible: 1
is_isothermal: 1
is_newtonian: 0
is_turbulent: 0
type_body_force: 0
type_non_newtonian: 2
type_non_turbulent: 0 | is_compressible: 1
is_isothermal: 0
is_newtonian: 1
is_pressure_work: 1
is_thermal_conductive: 1
is_turbulent: 0
is_viscosity_diffusion: 1
type_body_force: 1
type_mu_temperature: 0
type_newtonian: 1
type_non_turbulent: 0 |
| SINDy +GPT (with PhysPDE prompt) | is_compressible: 1
is_isothermal: 1
is_newtonian: 0
is_turbulent: 1
type_body_force: 0
type_non_newtonian: 3
type_turbulent: 1 | is_compressible: 1
is_isothermal: 1
is_newtonian: 0
is_turbulent: 1
type_body_force: 0
type_non_newtonian: 2
type_turbulent: 1 | is_compressible: 1
is_isothermal: 0
is_newtonian: 1
is_pressure_work: 1
is_thermal_conductive: 1
is_turbulent: 1
is_viscosity_diffusion: 0
type_body_force: 1
type_mu_temperature: 0
type_newtonian: 2
type_turbulent: 0 |
| GT+ GPT (with PhysPDE prompt) | is_compressible: 1
is_isothermal: 1
is_newtonian: 0
is_turbulent: 0
type_body_force: 0
type_non_newtonian: 3
type_non_turbulent: 0 | is_compressible: 1
is_isothermal: 1
is_newtonian: 0
is_turbulent: 0
type_body_force: 0
type_non_newtonian: 2
type_non_turbulent: 1 | is_compressible: 1
is_isothermal: 0
is_newtonian: 1
is_pressure_work: 1
is_thermal_conductive: 1
is_turbulent: 0
is_viscosity_diffusion: 1
type_body_force: 1
type_mu_temperature: 0
type_newtonian: 1
type_non_turbulent: 0 |

of wall-time (parallelized across 128 CPU cores). We randomly selected one decision combination as the ground truth and divided the dataset along the temporal axis into training, validation, and test sets. To simulate data noise, we applied a factor of $1 + 0.01 * \mathcal{N}(0, 1)$ to the training and validation datasets. The results of the hypothesis selection (RQ1) are shown in Table.16. According to the table, our method identifies the correct hypotheses in $37\%$ of the wall-time required by Exhaustive Search (ES), whereas ES-TL fails to produce the correct result.

Despite the problem's small search space, the simulation high cost makes exhaustive search impractical. Previously, searches were manually conducted, relying on expert experience. Our method offers an efficient and automated way to explore the search space, facilitating scientific discovery.

## 6 CONCLUSION

In this paper, we have proposed PhysPDE, a new ML4Sci task, with two datasets and baseline algorithms. It offers a unified framework for discovering and interpreting PDEs, bridging the gap between theoretical understanding and practice in scientific research.

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

Table 11: Fluid Mechanics Hypothesis Decision Forest

| Decision ID | Decision Name | Value | Hypothesis Name | Child Decision Name |
|---|---|---|---|---|
| 0 | is_turbulent | 0
1 | non_turbulent
turbulent | type_non_turbulent
type_turbulent |
| 1 | type_non_turbulent | 0
1 | laminar
creeping | -
- |
| 2 | type_turbulent | 0
1 | k-epsilon
realizable k-epsilon | -
- |
| 3 | is_newtonian | 0
1 | non_newtonian
newtonian | type_non_newtonian
type_newtonian |
| 4 | type_non_newtonian | 0
1
2 | powerlaw(mu_app)
carreau
new_non_newtonian_1 | is_dilatant
-
poly_order
Fourier_order |
| 5 | poly_order | 0
1
2 | poly_order_0
poly_order_1
poly_order_2 | -
-
- |
| 6 | Fourier_order | 0
1
2 | Fourier_order_0
Fourier_order_1
Fourier_order_2 | -
-
- |
| 7 | is_dilatant | 0
1 | pseudoplastic
dilatant | -
- |
| 8 | type_newtonian | 0
1 | inviscid
newtonian | -
- |
| 9 | is_isothermal | 0
1 | nonisothermal
isothermal | type_mu_temperature
is_thermal_conductive
is_thermal_conductive
is_pressure_work
is_viscosity_diffusion
- |
| 10 | type_mu_temperature | 0
1
2
3 | mu_temperature_independent
powerlaw(mu_T)
Sutherland
Andrade | -
-
-
- |
| 11 | is_thermal_conductive | 0
1 | not_thermal_conductive
Fourier | -
- |
| 12 | is_pressure_work | 0
1 | no_pressure_work
pressure_work | -
- |
| 13 | is_viscosity_diffusion | 0
1 | no_viscosity_diffusion
viscosity_diffusion | -
- |
| 14 | is_compressible | 0
1 | incompressible
compressible | -
- |
| 15 | type_body_force | 0
1 | no_body_force
gravity | -
- |

## A  FLUID MECHANICS HYPOTHESIS DEFINITION

In this paper, we focus on fluid mechanics. The full decisions, hypotheses, and parent-child relations are listed in Table 11.

## B  FLUID MECHANICS NOTATION DEFINITION

$$\text{Material Derivative} \quad \frac{\mathrm{D}f}{\mathrm{D}t} = \frac{\partial f}{\partial t} + \mathbf{u} \cdot \nabla f$$

$$\text{Pressure Work} \quad Q_p = \beta T(\frac{\partial p}{\partial t} + \mathbf{u} \cdot \nabla p)$$

$$\text{Viscosity Diffusion} \quad Q_{vd} = \boldsymbol{\tau} : \mathbf{u}$$

$$\text{Body Force} \quad \mathbf{F} = (0, -\rho g)^T$$

$$\text{Deviatoric Stress} \quad \boldsymbol{\tau} = \mu(\nabla \mathbf{u} + (\nabla \mathbf{u})^T) - \frac{2}{3}\mu(\nabla \cdot \mathbf{u})\mathbf{I}$$

$$\text{Shear Rate} \quad \gamma = \sqrt{2\mathbf{S} : \mathbf{S}}$$

$$\text{Strain Rate} \quad \mathbf{S} = \frac{1}{2}(\nabla \mathbf{u} + (\nabla \mathbf{u})^T)$$

## C  FLUID MECHANICS BASIC INTRODUCTION

Fluid dynamics is governed by the principles of conservation of mass, momentum, and energy, which are mathematically represented by the continuity equation, the Navier–Stokes equations, and the energy equation, respectively.

### C.1  CONTINUITY EQUATION (CONSERVATION OF MASS)

This equation ensures that mass is neither created nor destroyed within a fluid system. For compressible fluid, it is expressed as:

$$\frac{\partial \rho}{\partial t} + \nabla \cdot (\rho \mathbf{u}) = 0 \tag{5}$$

where $\mathbf{u}$ is the fluid velocity vector, and $\rho$ is density. An incompressible fluid has constant density, thus $\frac{\partial \rho}{\partial t} = 0$ and $\nabla \rho = 0$, and the continuity equation reduces to:

$$\nabla \cdot \mathbf{u} = 0 \tag{6}$$

### C.2  NAVIER–STOKES EQUATION (CONSERVATION OF MOMENTUM)

These equations describe the motion of fluid substances by accounting for forces acting on the fluid, including pressure, viscous forces, and external forces. They are formulated as:

$$\rho \frac{\partial \mathbf{u}}{\partial t} + \rho(\mathbf{u} \cdot \nabla)\mathbf{u} = \nabla \cdot [-p\mathbf{I} + \tau] + \mathbf{F} \tag{7}$$

where $p$ is the pressure, $\mathbf{F}$ represents external body forces (e.g., gravity). $\tau$ is the deviatoric stress tensor:

$$\tau = \mu \left(\nabla \mathbf{u} + (\nabla \mathbf{u})^\top\right) - \frac{2}{3}\mu(\nabla \cdot \mathbf{u})\mathbf{I}. \tag{8}$$

The $\mu$ is the dynamic viscosity. For Newtonian fluids, the dynamic viscosity is a constant. The most commonly encountered fluids are Newtonian, such as water, gases, and ethanol. However, there are a number of fluids that do not have constant viscosity, known as non-Newtonian. Examples of non-Newtonian fluids include blood, paint, and cosmetic products.

### C.3  ENERGY EQUATION (CONSERVATION OF ENERGY)

This equation accounts for the energy (temperature) changes within the fluid due to conduction, convection, and internal energy variations. It is given by:

$$\rho C_p(\frac{\partial T}{\partial t} + \mathbf{u} \cdot \nabla T) = \nabla \cdot (k\nabla T) + Q_p + Q_{vd} \tag{9}$$

Table 12: Decisions in S1 (incorrect decisions colored red)

|  | Top-1 | Top-2 | Top3 |
|---|---|---|---|
| Ours | is_compressible: 1
is_isothermal: 1
is_newtonian: 1
is_turbulent: 0
type_body_force: 0
type_newtonian: 1
type_non_turbulent: 0 | is_compressible: 0
is_isothermal: 1
is_newtonian: 1
is_turbulent: 0
type_body_force: 0
type_newtonian: 1
type_non_turbulent: 0 | is_compressible: 1
is_isothermal: 1
is_newtonian: 0
is_turbulent: 0
type_body_force: 0
type_non_newtonian: 1
type_non_turbulent: 0 |
| ES | is_compressible: 1
is_isothermal: 1
is_newtonian: 1
is_turbulent: 0
type_body_force: 0
type_newtonian: 1
type_non_turbulent: 0 | is_compressible: 1
is_isothermal: 0
is_newtonian: 1
is_pressure_work: 0
is_thermal_conductive: 0
is_turbulent: 0
is_viscosity_diffusion: 0
type_body_force: 0
type_mu_temperature: 0
type_newtonian: 1
type_non_turbulent: 0 | is_compressible: 1
is_isothermal: 0
is_newtonian: 1
is_pressure_work: 0
is_thermal_conductive: 1
is_turbulent: 0
is_viscosity_diffusion: 0
type_body_force: 0
type_mu_temperature: 0
type_newtonian: 1
type_non_turbulent: 0 |

where $C_p$ is the specific heat capacity at constant pressure, $T$ is the temperature, $k$ is the thermal conductivity, $Q_p, Q_{vd}$ represents pressure work and viscosity diffusion in turbulence. For isothermal fluid, the temperature is assumed constant, thus the above equation is constantly zero.

These fundamental equations collectively describe fluid behavior under various conditions and are essential for analyzing and predicting fluid flow phenomena.

## D   FLUID MECHANICS HYPOTHESIS SELECTION RESULTS

The learned decisions and values of all scenarios are listed in Table 12-14.

## E   ALGORITHM PRELIMINARIES

Our search algorithm is inspired by two established algorithms MCTS and Sequential Threshold Ridge Regression (STRidge)

**MCTS** The MCTS algorithm (Coulom, 2006) stands out as a robust approach frequently utilized in solving optimal decision-making problems. This method employs a tree search mechanism that adeptly balances the dual aspects of exploration and exploitation. At its core, the MCTS algorithm unfolds through a cyclical iteration of four fundamental steps.

**1) Selection**. The MCTS agent, starting from the root node, selects successive child nodes according to a given policy until an expandable node or a leaf node is encountered.

**2) Expansion**. The search tree is then expanded by selecting an unvisited child at an expandable node.

**3) Simulation**. Following expansion, the agent performs independent simulations from the selected child node, executing random actions. The simulation halts if the agent reaches a terminal state or a predefined time constraint.

**4) Backpropagation**. Statistics of nodes along the random path are updated according to the ultimate search result.

To balance exploration and exploitation, the Upper Confidence Bounds applied for Trees (UCT) algorithm is integrated (Kocsis & Szepesvári, 2006). During the selection phase, child node with the

Table 13: Decisions in S2 (incorrect decisions colored red)

| | Top-1 | Top-2 | Top3 |
|---|---|---|---|
| Ours | is_compressible: 1
is_isothermal: 1
is_newtonian: 0
is_turbulent: 0
type_body_force: 0
type_non_newtonian: 3
type_non_turbulent: 0 | is_compressible: 0
is_isothermal: 1
is_newtonian: 0
is_turbulent: 0
type_body_force: 0
type_non_newtonian: 3
type_non_turbulent: 0 | is_compressible: 1
is_isothermal: 0
is_newtonian: 0
is_pressure_work: 0
is_thermal_conductive: 1
is_turbulent: 0
is_viscosity_diffusion: 0
type_body_force: 0
type_mu_temperature: 0
type_non_newtonian: 3
type_non_turbulent: 0 |
| ES | is_compressible: 1
is_isothermal: 1
is_newtonian: 0
is_turbulent: 0
type_body_force: 0
type_non_newtonian: 3
type_non_turbulent: 0 | is_compressible: 1
is_isothermal: 0
is_newtonian: 0
is_pressure_work: 1
is_thermal_conductive: 1
is_turbulent: 0
is_viscosity_diffusion: 0
type_body_force: 0
type_mu_temperature: 0
type_non_newtonian: 3
type_non_turbulent: 0 | is_compressible: 1
is_isothermal: 0
is_newtonian: 0
is_pressure_work: 1
is_thermal_conductive: 0
is_turbulent: 0
is_viscosity_diffusion: 0
type_body_force: 0
type_mu_temperature: 0
type_non_newtonian: 3
type_non_turbulent: 0 |

Table 14: Decisions in S3 (incorrect decisions colored red)

| | Top-1 | Top-2 | Top3 |
|---|---|---|---|
| Ours | is_compressible: 1
is_isothermal: 0
is_newtonian: 1
is_pressure_work: 1
is_thermal_conductive: 1
is_turbulent: 0
is_viscosity_diffusion: 1
type_body_force: 1
type_mu_temperature: 0
type_newtonian: 1
type_non_turbulent: 0 | is_compressible: 1
is_isothermal: 0
is_newtonian: 1
is_pressure_work: 0
is_thermal_conductive: 1
is_turbulent: 0
is_viscosity_diffusion: 1
type_body_force: 1
type_mu_temperature: 0
type_newtonian: 1
type_non_turbulent: 0 | is_compressible: 1
is_isothermal: 0
is_newtonian: 1
is_pressure_work: 0
is_thermal_conductive: 1
is_turbulent: 0
is_viscosity_diffusion: 0
type_body_force: 1
type_mu_temperature: 3
type_newtonian: 1
type_non_turbulent: 0 |
| ES | is_compressible: 1
is_isothermal: 0
is_newtonian: 1
is_pressure_work: 0
is_thermal_conductive: 1
is_turbulent: 0
is_viscosity_diffusion: 0
type_body_force: 1
type_mu_temperature: 0
type_newtonian: 1
type_non_turbulent: 0 | is_compressible: 1
is_isothermal: 0
is_newtonian: 1
is_pressure_work: 0
is_thermal_conductive: 1
is_turbulent: 0
is_viscosity_diffusion: 1
type_body_force: 1
type_mu_temperature: 0
type_newtonian: 1
type_non_turbulent: 0 | is_compressible: 0
is_isothermal: 0
is_newtonian: 1
is_pressure_work: 0
is_thermal_conductive: 1
is_turbulent: 0
is_viscosity_diffusion: 0
type_body_force: 1
type_mu_temperature: 0
type_newtonian: 1
type_non_turbulent: 0 |

maximal UCT value is selected:

$$\text{UCT}(s, a) = Q(s, a) + c\sqrt{ln[N(s)]/N(s, a)}, \tag{10}$$

where $Q(s, a)$ is the average win ratio of playing action a in state s in previous simulations; $N(s)$ is the number of times state s has been visited; $N(s, a)$ is the number of times action a gets selected at state s. Constant $c$ controls the balance between exploration and exploitation, which is represented by $\sqrt{ln[N(s)]/N(s, a)}$ and $Q(s, a)$ respectively. The convergence and efficiency of this algorithm have been rigorously validated as demonstrated by Shah et al. (2020).

**STRidge.** STRidge (Rudy et al., 2017) is an advanced algorithm that combines concepts from both ridge regression and sparsity-promoting techniques. It is composed of an iterative process of ridge regression and thresholding. As of the ridge regression step, penalty terms are incorporated into the objective function to limit the number of predictor variables considered relevant:

$$\hat{\xi} = \arg\min_{\xi} ||\mathbf{\Theta}(\mathbf{U})\xi - \mathbf{U}_t||_2^2 + \epsilon||\xi||_2^2, \tag{11}$$

where $\mathbf{\Theta}$ denotes the library of candidate terms, $\mathbf{U}$ is data, and $\epsilon$ regularization coefficient.

After the initial ridge regression step, STRidge applies a thresholding mechanism, eliminating terms with coefficients below a threshold value. The idea is to eliminate predictors that have a negligible effect on the target variable, thereby promoting sparsity in the model. This process continues until it converges to a stable set of predictors or until a specified number of iterations is reached. STRidge poses stricter regularization and penalties for overfitting, therefore achieving outstanding performance in PDE discovery tasks.

## F  HSTS AND STOP ALGORITHM

The main algorithm is summarized in Alg. 1, and we will elaborate on its details below.

---

**Algorithm 1:** HSTS+STOP Algorithm

---

**Input**    : Decision Forest, Operator Library, Observed Data
**Param**  : (HSTS) loss bound $\ell_{\text{max,min}}$, risk fraction $\epsilon$, max episode *ME*, number of rollout *NR*, exploration weight
**Param**  : (STOP) Regulariazation $\boldsymbol{\lambda}_*$, threshold $T$, max iteration
**Output** : Optimal decision $h$, coefficients $\theta_h$
**Output** : (Optionally) new expression $m$, coefficients $\theta_m$

1 Initialize $h$ as empty vector, initialize an empty cache;
2 **for** *each episode* $\leq$ ME **do**
3     **if** *h has no child* **then**
4        break;
5     **for** *each rollout* $\leq$ NR **do**
6        **Selection**: $h_0 \longleftarrow$ UCT selection of object $J$(Eq. 14) among descendants of $h$;
7        **Expansion**: $h_0 \longleftarrow$ Repeatedly select a child of $h_0$ until no child available;
8        **Simulation**: Calculate reward(Eq.12) of $h_0$ by solving Eq.15 via STOP;
9        **Backpropagation**: Propagate and cache the reward;
10     **Action**: $h \longleftarrow$ the child maximizing $J$(Eq. 14);
11 **Return** best solution in the cache;

---

### F.1  HYPOTHESES SELECTION TREE SEARCH (HSTS)

This section addresses solving the outer-level decision vector $h$. Although the decision forest effectively represents various physics models and constraints, the search space is vast, making naive enumeration impractical. With $n$ nodes in the forest and a minimum of 2 choices per node, the total possible decision instances are $\mathcal{O}(2^n)$, excluding interdependencies. To tackle this challenge, we employ the Monte Carlo Tree Search (MCTS) algorithm to efficiently explore the search space for our hypothesis selection task.

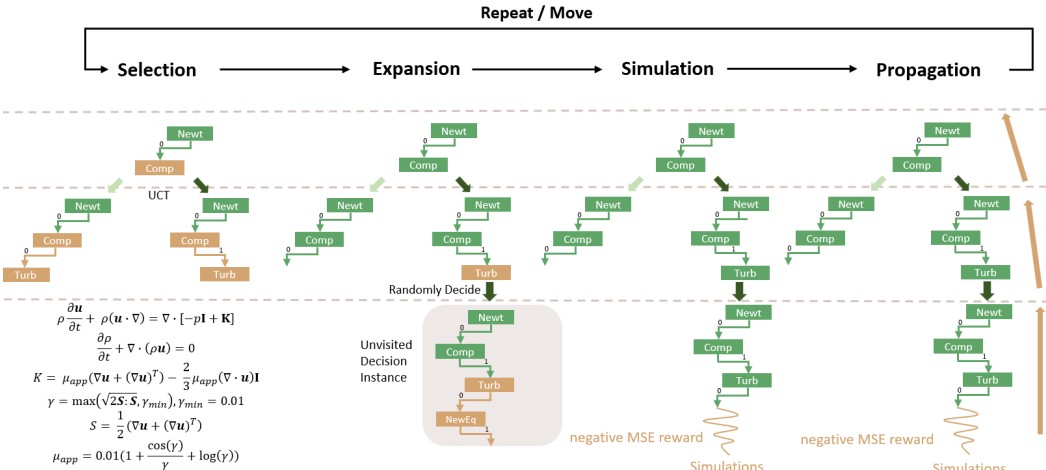

Figure 4: Overview of the proposed Hypothesis Selection Tree Search (HSTS) algorithm.

MCTS is a heuristic search algorithm renowned for simplicity and effectiveness in complex decision-making scenarios (Silver et al., 2017). It navigates the search space by expanding selected search nodes and back-propagating simulation results through the tree structure. In this paper, the MCTS algorithm is adapted to align with the unique requirements of Hypothesis Selection. The adaptation involves customizing the search node, reward, and action objective of MCTS to handle the decision forest representation and evaluation of PDEs within the context of Hypothesis Selection. The resulting algorithm is thus named HSTS. Figure 4 demonstrates an illustrative example.

**Search Node.** In MCTS, a search node represents an (in)complete decision instance from the decision forest, essentially a set of tree paths from roots. The search begins with an empty decision node and progresses to a terminal node representing a complete decision. A child node is generated by extending the parent's decision instance with the next decision, adhering to the predefined sequential order of decisions within the forest.

**Reward Function.** Once a valid decision $\boldsymbol{h}$ is sampled, we construct a system of PDEs and evaluate the system with a reward function $R$ on observation data $X$. To simplify notation, we denote the minimal loss of decision $\boldsymbol{h}$ as $\ell(\boldsymbol{h})$:

$$\ell(\boldsymbol{h}) = \min_{\boldsymbol{\theta_h}} \min_{m, \boldsymbol{\theta_m}} \ell(X, Eq), \quad C \geq 0. \tag{12}$$

While a straightforward reward definition could be the negative of $\ell(\boldsymbol{h})$, the unpredictable numerical error level may lead to an unbounded reward range. This unpredictability can potentially disrupt the balance between exploration and exploitation in the UCT selection. Thus, we normalize the loss and constrain it within the range $[0, 1]$:

$$R(\boldsymbol{h}) = 1 - \frac{\log \ell(\boldsymbol{h}) - \log \ell_{\min}}{\log \ell_{\max} - \log \ell_{\min}}, \tag{13}$$

where the hyper-parameters $\ell_{\max}$ and $\ell_{\min}$ serve as approximate bounds for the loss.

**Risk-seeking Objective.** After every $N$ back-propagation iteration, MCTS either terminates or selects an action for execution, maximizing the current state's objective. Actions correspond to moving to child search nodes, while the state encompasses the current search node and all visiting and rewarding histories. In traditional Monte Carlo Tree Search (MCTS), the objective for action $a$ at state $s$ is the expected reward under policy $p$, denoted as $J(s, a) = \mathbb{E}_{\boldsymbol{h} \sim p(s, a)}[R(\boldsymbol{h})]$. This approach, however, does not align with the priorities in Hypothesis Selection, which emphasizes the performance of top outcomes over average-case scenarios. Similar challenges in policy-gradient methods led to the adoption of risk-seeking objectives, concentrating on the highest rewards within a batch. Inspired by such methods (Petersen et al., 2019; Tamar et al., 2014), we propose an $\epsilon$-parameterized risk-seeking objective for MCTS.

$$J(s, a; \epsilon) = \mathbb{E}_{\boldsymbol{h} \sim p(s, a)}[R(\boldsymbol{h}) | R(\boldsymbol{h}) \geq R_\epsilon(p(s, a))]. \tag{14}$$

This objective averages the rewards of the top $\epsilon$ fraction of samples from the distribution $p(s, a)$, disregarding all other samples falling beneath this performance threshold.

### F.2 SEQUENTIAL THRESHOLD OPTIMIZATION (STOP)

Here we address the challenge of determining the boolean vector $m$ as specified on the right-hand-side of Eq. 12. The objective is to select an optimal combination of functions from a predefined function library for constructing a new expression, where each element of vector $m$ denotes the inclusion (or exclusion) of a specific function. The loss function indicates that it is an $\mathcal{L}_0$ regularized sparse optimization problem, which is proved to be NP-hard (Virgolin & Pissis, 2022). An intuitive idea of this complexity is the exponential growth of the search space with the size of the function library, $n$, leading to $2^n$ possible function combinations.

Inspired by the sequential threshold ridge regression technique utilized in PDE discovery (Rudy et al., 2017), we propose a Sequential Threshold Optimization (STOP) method. It adopts an iterative backward selection strategy, initially setting all elements of $m$ to one. In each iteration, it tackles the following constrained optimization problem:

$$\boldsymbol{\theta}_h^*, \boldsymbol{\theta}_m^* = \operatorname*{argmin}_{\boldsymbol{\theta}_h, \boldsymbol{\theta}_m} \quad \ell(X, Eq) + \lambda_{\boldsymbol{\theta}_m} \|\boldsymbol{\theta}_m\|_2^2 \tag{15}$$

$$\text{s.t.} \quad \boldsymbol{\theta}_m(1 - m) = 0, \quad C \geq 0,$$

where $\lambda_{\boldsymbol{\theta}_m}$ serves as a hyperparameter to modulate the regularization strength. Subsequently, $m$ is updated to deactivate elements where $\boldsymbol{\theta}_m^*$ falls below a predefined threshold T, with updated rule $m = \mathbf{1}[\boldsymbol{\theta}_m^* > \text{T}]$. This iterative refinement continues until

$m$ stabilizes, effectively reducing the search space by excluding insignificant function choices.

The STOP terminates within at most $\text{size}(m)$ iterations since each iteration either diminishes the count of non-zero elements of $m$ or terminates the algorithm. This approach trades optimality for efficiency, with STOP's output not assured to be the optimal solution due to its inherent greedy selection mechanism. However, as demonstrated by our empirical findings in Section 4.4, STOP is capable of accurately identifying the correct expression with an appropriate configuration of hyperparameters.

### F.3 NUMERICAL SOLVER OF PDE INVERSE PROBLEM

To solve the PDE parameter estimation problem Eq. 15 or inverse problem (Isakov, 2006), we employ a two-step approach: 1) discretization of PDE derivatives leveraging the centered finite difference method (Strikwerda, 2004), transforming it into an ordinary constrained optimization problem, and 2) application of a sequential least squares quadratic programming solver for optimization (Stoer, 1985).

## G RESULT VISUALIZATION

The performance is visualized in Figure. 5-11.

## H LASER FUSION HYPOTHESIS DEFINITION AND RESULTS

## I DISCUSSION

A traditional benchmark should be a comprehensive evaluation of existing methods. We acknowledge that PhysPDE may not fully align with the traditional definition of a benchmark paper. However, in a broader sense, we believe that a benchmark typically includes the following components:

1) a dataset (e.g., Fluid Mechanics dataset);

2) a task setting (e.g., Physics Interpretation task);

3) a workflow for evaluation (e.g., integrated/pipelined approaches);

4) evaluation results.

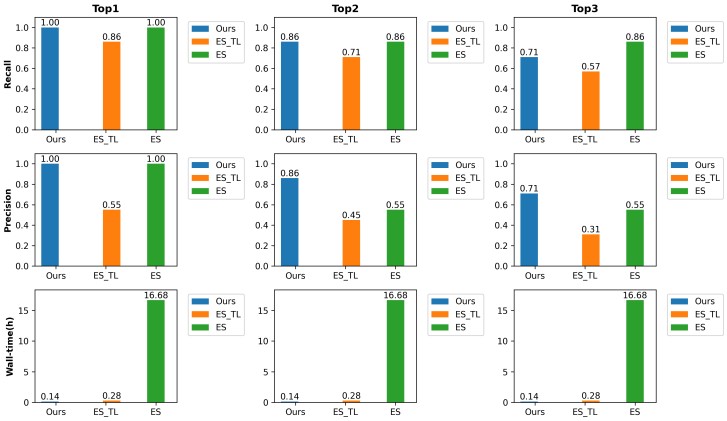

Figure 5: RQ1-S1

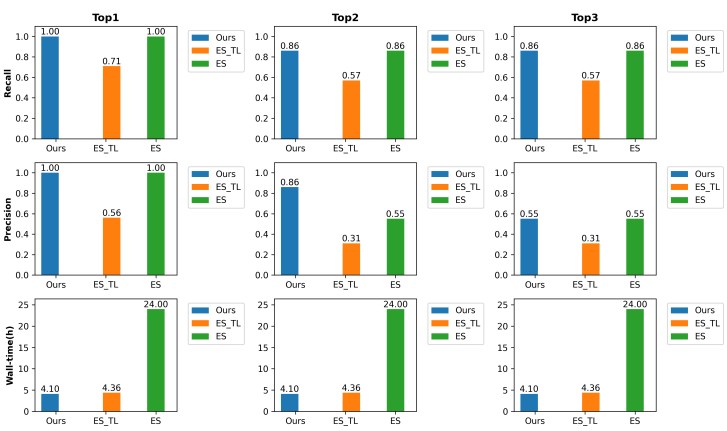

Figure 6: RQ1-S2

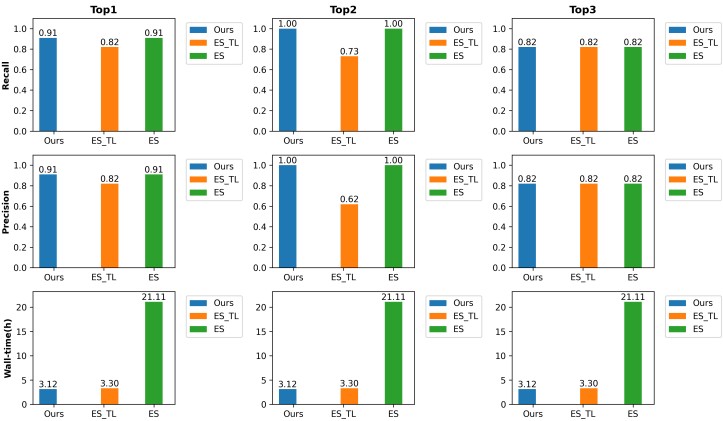

Figure 7: RQ1-S3

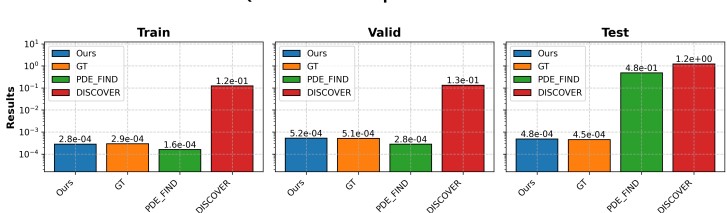

Figure 8: RQ2

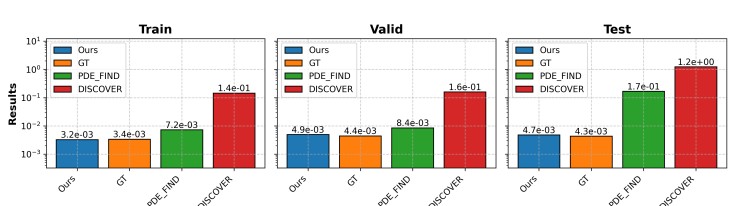

Figure 9: RQ3-S1

Figure 10: RQ3-S2

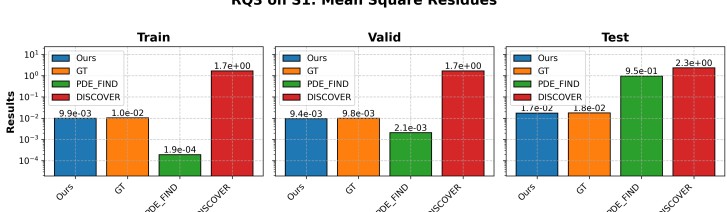

Figure 11: RQ3-S3

Table 15: Laser Fusion Hypothesis Decision Forest

| Decision ID | Decision Name | Value | Hypothesis Name | Child Decision Name |
|---|---|---|---|---|
| 0 | is_magnetic_field | 0 | no_magnetic_field | - |
| | | 1 | magnetic_field | - |
| 1 | range_divergence_angle | 0 | divergence_angle$<$45 | - |
| | | 1 | divergence_angle$\geq$45 | - |
| 2 | range_temperature | 0 | temperature$<$T_f | - |
| | | 1 | temperature$\geq$T_f | - |
| 3 | is_dopant | 0 | no_dopant | - |
| | | 1 | dopant | type_dopant |
| 4 | type_dopant | 0 | dopant_Au | - |
| | | 1 | dopant_Br | - |
| | | 2 | dopant_Cl | - |

Table 16: RQ1 on Laser Fusion: Hypothesis Selection.

| Model | | ↑ Recall | ↑ Precision | ↓ Wall-time(h) |
|---|---|---|---|---|
| | Top-1 | **1.00** | **1.00** | |
| Ours | Top-2 | 0.40 | 0.40 | **8.67** |
| | Top-3 | 0.80 | 0.80 | |
| | Top-1 | 0.80 | 0.80 | |
| ES-TL | Top-2 | 0.60 | 0.60 | 8.70 |
| | Top-3 | 0.80 | 0.80 | |
| | Top-1 | 0.60 | 0.60 | |
| ES | Top-2 | **0.80** | **0.80** | 23.21 |
| | Top-3 | **1.00** | **1.00** | |

Our paper's primary contributions lie in components 1, 2, and (the new algorithm implemented for) 3. The 4-th point is relatively weak since not enough baselines are available for our new task. However, we have demonstrated that the combination of the existing baseline (i.e. the SINDy+GPT pipeline) can be applied. In this sense, PhysPDE will provide a foundation for assessing and improving more ML4Sci algorithms in the future.

## J LIMITATIONS

It is important to acknowledge the current limitations, including scalability to exceedingly large datasets and the need for further refinement when dealing with highly complex PDE scenarios. Future endeavors will be directed toward addressing these challenges and exploring the application of PhysPDE to real-world data, enhancing its utility. A potential negative societal impact could be reducing the need for human expertise in certain scientific fields, leading to job losses.

