# OpenReview forum: "PhysPDE: Rethinking PDE Discovery and a Physical Hypothesis Selection Benchmark"
_ICLR.cc/2025/Conference — ICLR 2025 Poster_

### Official Review · Reviewer_A96C · 2024-10-20

**Soundness:** 4
**Presentation:** 3
**Contribution:** 3
**Rating:** 8
**Confidence:** 4

**Summary:**

Physical insights via following existing physical theories and hypotheses are benchmarked in this paper to learn PDE. This allows for possibly good interpolation and extrapolation of PDE learning.

**Strengths:**

The overall idea is interesting and useful. I like the idea and think it is going in the correct direction -- to make PDE learning physically interpretable via construction of differential equations under the current physical framework, as opposed to symbolic regression only. The datasets used are in different areas, and the physical laws considered are representative.

**Weaknesses:**

It still remains unclear to me how your decisions (such as is_compressible, is_newtonian) directly affect the PDE system. You have a Figure 3 that shows how the PDE formulation is defined under both different conditions, such as is_compressible=yes and is_newtonian=false. However, I still do not know how this is constructed. What is the connection between these conditions and the PDE appearance? What does it mean for a PDE to have is_compressible=yes? How does that reveal in the PDE terms? I know you are using a decision tree algorithm and use data-label pairs to train a model to predict whether is_compressible=yes. However, but as a reader, understanding what does it mean for a PDE to be is_compressible=yes is important for them to know whether and how that is learnable. This is also good introduction for ML people who may be less familiar with physics.

**Questions:**

I hope to see the weaknesses being addressed. It should not be hard to address them since these are clarity issues, but they are critical.

---

> ### Author Response · Authors · 2024-11-23
>
> We sincerely thank the reviewer for the insightful and encouraging feedback. We are delighted that the reviewer finds our approach of leveraging existing physical theories and hypotheses to benchmark PDE learning both interesting and useful. Below, we address the reviewer’s comment in detail.
>
> ---
> **W1. How your decisions (such as is_compressible, is_newtonian) directly affect the PDE system?**
>
> **A1.** We have updated the Figure 3 ([link](https://postimg.cc/230gnr81)) to include more details. We also added an introduction to fluid mechanics in the Appendix. C, at Page. 15 of the draft ([link](https://openreview.net/pdf?id=G3CpBCQwNh)), including the conservation laws, compressible flow, and Newtonian fluids, and how they together determine the PDE system. Briefly speaking, 'compressible' means that fluid density is not constant, and 'Newtonian' means that dynamic viscosity is constant. For a more systematic introduction, please refer to the fluid mechanics textbook e.g. [1].
>
> Reference:
>
> [1] White, Frank M., and Henry Xue. Fluid mechanics. Vol. 3. New York: McGraw-hill, 2003.

---

> > ### Comment · Reviewer_A96C · 2024-11-25
> >
> > Thanks for the update. It is helpful, but the core question remains unanswered. What is the connection between properties like "is_compressible=yes" and PDE terms? I understand when some PDE terms come together, they may have is_compressible=yes, but how would you do the reverse? That is, if you have is_compressible=yes, how do you constrain the model to only search in equations that are "is_compressible=yes"? How do you do that?

---

> ### Author Response · Authors · 2024-11-25
>
> Thank you for the response, as for your additional questions, we clarify as follows:
>
> ---
> >1.  How to derive PDE terms given a hypothesis?
>
> We start from the full PDE system of the unchangeable physics theorem (e.g. conservation laws in fluid mechanics). On top of that, the derivation is done by plugging the expression of the selected hypothesis into the PDE of conservation laws.  All the symbolic calculations are done automatically by a computer algebra system (CAS) library.
>
> For example, the mass conservation is $\frac{\partial \rho}{\partial t}+\nabla \cdot(\rho \mathbf{u})=0$. If one applies the compressible hypothesis, it does not change, but if one applies the incompressible hypothesis, it becomes $\nabla \cdot \mathbf{u} = 0 $.
>
> ---
> >2.  How do you constrain the model to only search in equations that are is_compressible=yes?
>
> In the hypothesis search (outer-level optimization), the algorithm can only modify the hypothesis, and can not modify the equation directly. In addition, the hypothesis search is a tree-based algorithm, thus once a compressible hypothesis is selected, then all descendant searches are compressible.
>
> In the new equation discovery (inner-level optimization), the algorithm can modify the equation directly but is confined only to the specific part of the equation allowed by the specific hypothesis.
>
> For example, in our S2 experiment, the new equation discovery is allowed to discover a new expression of viscosity $\mu$ (based on the non-newtonian hypothesis), but is not allowed to modify the compressible hypothesis and its corresponding expression about $\rho$. In our current setting, no new expressions are related to compressibility. Thus the inner-level search will not change compressibility.
>
> ----
>
> Thank you once again for your support and contributions to this review process. If there are any remaining questions or clarifications needed from us, we would be more than happy to assist promptly.

---

> > ### Comment · Reviewer_A96C · 2024-11-26
> >
> > Thanks for clarifying. This is helpful. Please make sure this is included in the main paper.

---

> > > ### Author Response · Authors · 2024-11-27
> > > **Official Comment by Authors**
> > >
> > > Dear Reviewer A96C,
> > >
> > > We hope this message finds you well. We sincerely thank you for taking the time to review our work and for your thoughtful comments. We have carefully revised the main paper according to your suggestions. We are especially grateful for your decision to increase the score, and we truly appreciate the constructive feedback that led to this improvement.
> > >
> > > Thank you once again for your valuable support and thoughtful contributions to this review process. Please let us know if there are any new issues or further suggestions.
> > >
> > > Best regards,
> > >
> > > PhysPDE Authors

---

### Official Review · Reviewer_Xb5C · 2024-11-02

**Soundness:** 2
**Presentation:** 2
**Contribution:** 2
**Rating:** 6
**Confidence:** 2

**Summary:**

The paper proposes a new ML4Sci task which focuses on identifying the hypothesis that best align with the data. Prior physics hypothesis and theories are used and the proposed paradigm focuses on identifying which ones align best with the data. The approach is introduced using a mixed-integer programming formulation. Furthermore, the paper designs new data sets for this problem, specifically from the domains of Fluid Mechanics and Laser Fusion.

**Strengths:**

Overall, the paper proposes a novel task in the ML4Sci field, which could potentially lead to new methods and approaches developed in this field.

**Weaknesses:**

- Unless I missed it, the data introduced data sets are not open access and aren't shared during the review process, which limits the possible impact of the paper.
- Given the complexity of the proposed approach (Fig. 1 which includes many components) and newly generated data sets, the results of the paper are not reproducible with the current information available to the reader.

**Questions:**

1) Are you planning to release the data sets and the code? (Are these available for review?)
2) Could you elaborate more on the use of LLMs and in particular GPT-4o for extracting physics hypothesis from the data? Why are LLMs specifically appropriate for this task?

Update after discussion period: I have increased my score while keeping my confidence low. The main reproducibility concern has been addressed during the review process.

---

> ### Author Response · Authors · 2024-11-23
>
> We would like to express our sincere gratitude to the reviewer for thoroughly evaluating our paper and providing constructive feedback. We are pleased that the reviewer recognizes the novelty of our proposed ML4Sci task and the significance of new datasets. Below, we address the reviewer's comments and provide detailed clarifications and responses.
>
>
> ---
>
> **Q1: the release of the dataset and code**
>
> **A1.** In the original submission, we have provided the Fluid dataset link in the supplementary material. Here, we release both the dataset ([link](https://zenodo.org/records/11530771?token=eyJhbGciOiJIUzUxMiJ9.eyJpZCI6IjgwZGFkMTJiLTY2NjYtNGY0MS04YzI4LTZkMzRjNmM2ZGRlZCIsImRhdGEiOnt9LCJyYW5kb20iOiI0NTA4MWU5MzkxOGU4YjYwNjdlMGJkYmUzY2NmYjM5YSJ9.50vo70qCuAfIokz6KsUps-DaQbppGM75joD8DpyLi-6lVn3DGgtTDzv6MSgRx2wl9RmTi8T1yjx785gHJuEyvA)) and the code ([link](https://zenodo.org/records/14163760?token=eyJhbGciOiJIUzUxMiJ9.eyJpZCI6IjYyYjgwMjExLTU4NTItNGY3Yy1hNDRmLTZjMzRkYTU2ZDEyYyIsImRhdGEiOnt9LCJyYW5kb20iOiI0ZGU5NDUzZGIyODZiZWU3MjZiZjZmMDllNTM3ZjA1NiJ9.P9GKJpCiR9i18IITsCdrXMOpqOweLt0IMvH87lI4AyeadF67jDgBAQ7loMIVaGNIVREHm0St20yvbtjCOkAC5A)) as required.  We genuinely hope that the source code can help you better understand our work.
>
>
> ---
> **Q2. justification of the use of LLMs in baseline**
>
>
> **A2.** Selecting physics hypothesis is a brand new ML4Sci task without suitable baseline method. Aside from recovering the symbolic representation, this task further requires the identification of physics properties corresponding to the data. Existing PDE discovery algorithms could only yield symbolic equations and fail to identify physical properties.
>
> To bridge this gap, we take a step back and seek an alternative model that can process both the equation (in math symbols) and hypothesis (in natural language), and can take in hypothesis forest prior knowledge as well. In our experiment, LLMs satisfy these conditions, as they manage to partially select the physics hypothesis given ground truth PDE and the hypothesis forest. Therefore, the pipeline of a classical PDE discovery method and an LLM is a suitable baseline for our new task. We chose GPT-4o because it has the strongest performance in our pre-experiment, among all LLMs accessible to us.
>
>
>
> ---
> **Claim of contribution and significance**
>
> Our major contribution is a new ML4Sci task on interpreting scientific data by physics hypothesis. Compared to previous math-only PDE discovery datasets, our PDE hypothesis dataset is additionally labeled with the PDE's physical properties such as compressibility. In addition, a novel algorithm framework is developed for this new task. This algorithm can identify both existing hypotheses and unknown expressions, potentially providing physics insights for scientists and enhancing the automation of physics discovery. Both the datasets and algorithms are publicly available.
>
> Reviewers have recognized our work as **"proposing a novel task"**(Xb5C), **"addressing the crucial challenge of physical interpretability"**(QbPL), **"interesting and useful"**(A96C) and **"well motivated"**(QC1C).

---

> > ### Author Response · Authors · 2024-11-25
> > **Follow-up on Review Feedback**
> >
> > Dear Xb5C,
> >
> > We hope this message finds you well. We are reaching out to kindly follow up on your review of PhysPDE. We greatly value your thoughtful feedback and insights, and we wanted to check if there are any additional clarifications or information you might need from us to assist in finalizing your review.
> >
> > If it is convenient, we would also appreciate it if you could provide an updated score based on our responses to your comments. Your assessment is highly important in the decision-making process, and we are happy to address any further concerns to ensure your confidence in the manuscript.
> >
> > Thank you so much for your time and effort in reviewing our work. Please don’t hesitate to reach out if there’s anything we can assist with.
> >
> > Best regards,
> >
> > PhysPDE Authors

---

> > > ### Comment · Reviewer_Xb5C · 2024-11-28
> > >
> > > Dear authors,
> > >
> > > Thanks for your reply. Unfortunately, both the data sets and the code have restricted access, so I don't seem to have an opportunity to review them.

---

> ### Author Response · Authors · 2024-11-26
> **Response Reminder**
>
> As we near the end of the discussion period, we would like to thank the reviewer again for the comments and provide a gentle reminder that we have posted a response to your comments. May we please check if our responses have addressed your concerns and improved your evaluation of our paper?

---

> ### Author Response · Authors · 2024-11-27
> **Response Reminder**
>
> As the discussion period comes to a close, we would like to once again express our sincere thanks for your valuable feedback. We would also like to kindly remind you that we have submitted a response to your comments, updating the data and code links as you requested. Could you please confirm if our responses have addressed your concerns and if they have had any impact on your assessment of our paper?

---

> ### Author Response · Authors · 2024-11-28
> **Data Accessibility**
>
> We are sorry for the misleading 'restricted' mark. The webpage for the data and code is indeed marked as 'restricted,' but it is available by accessing it through the link we provided. You just need to click 'download' to retrieve the files, as shown in [Fig1](https://postimg.cc/GBC2bt5m), [Fig2](https://postimg.cc/PCZJGYK7). We tested it on multiple computers, operator systems, and web browsers, and the download worked each time. Additionally, in case you are still unable to download it, we have uploaded the code and the training set of the S1 dataset in the supplementary materials. Due to the space limitations for supplementary materials, we could not upload the entire dataset.

---

> ### Author Response · Authors · 2024-11-29
> **Response Reminder**
>
> We would like to kindly ask if you were able to successfully download the dataset link we provided. If you still encounter any issues accessing it, please do not hesitate to inform us, and we will ensure the link is fully functional.
>
> Additionally, we would appreciate your insights on whether the supplementary materials we submitted have been helpful in evaluating our work. Your feedback is invaluable to us as we strive to improve the quality of our research.
>
> Thank you again for your time and effort in reviewing our manuscript.

---

> ### Author Response · Authors · 2024-12-01
> **Response Reminder**
>
> As the discussion period is coming to a close (only **2** days remaining), we would like to sincerely thank you again for your valuable feedback. We have posted a response to your data availability comments and would appreciate it if you could kindly confirm whether our clarifications have addressed your concerns and helped improve your evaluation of our paper.

---

> ### Public Comment · ~Min_Zhu1 · 2024-12-01
>
> I can access the dataset and code.

---

> > ### Author Response · Authors · 2024-12-02
> >
> > Thank you so much for your helpful public comment confirming the availability of our data and code. We greatly appreciate your assistance in ensuring transparency regarding the resources.

---

> ### Author Response · Authors · 2024-12-02
> **Response Reminder**
>
> As the discussion period is drawing to a close (with less than **1 day** remaining), we would like to sincerely thank you once again for your valuable feedback. We have provided a detailed response to your comments regarding data availability, and we would greatly appreciate it if you could kindly confirm whether our clarifications have addressed your concerns and contributed to a better evaluation of our paper. Thank you again for your time and consideration.

---

> ### Author Response · Authors · 2024-12-03
> **Final Response Reminder**
>
> As the discussion period is nearing its end (with less than **2 hours** remaining), we would like to express our gratitude once again for your feedback. We have provided a detailed response to your comments regarding **data availability**, and we would be appreciative if you could confirm whether our clarifications have addressed your concerns.

---

> ### Author Response · Authors · 2024-12-04
>
> We would like to sincerely thank the reviewers for deciding to increase the score of our manuscript. We are very glad that we were able to address your concerns and appreciate your feedback, which has contributed to the improvement of our work.

---

### Official Review · Reviewer_QbPL · 2024-11-03

**Soundness:** 3
**Presentation:** 3
**Contribution:** 3
**Rating:** 6
**Confidence:** 4

**Summary:**

This work addresses limitations in existing methods for discovering Partial Differential Equations (PDEs) from experimental data, particularly the lack of physical interpretability in machine-learning-driven approaches. This approach integrates machine learning with physical hypotheses and theories to guide the discovery of PDEs, prioritizing physical interpretability over purely mathematical accuracy.

Main contributions of this work are:

- The proposal of a new task, termed "PDE Interpretation", which involves selecting physical hypotheses (laws) that align with observed data rather than simply generating symbolic expressions, addressing the gap between mathematical representations and physical meaning.

- The authors, then, recast the discovery/solution task as a mixed-integer programming (MIP)-based approach formulated as a bi-level optimization problem. The outer layer selects the best-fitting physical hypotheses, while the inner layer focuses on expression recovery, involving symbolic regression when new expressions are necessary. They introduce a Monte Carlo Tree Search-inspired Hypothesis Selection Tree Search (HSTS) and a Sequential Threshold Optimization (STOP) algorithm for efficient problem-solving.

- Finally, the study provides two new datasets in fluid mechanics and laser fusion, designed to test the proposed framework's effectiveness in interpreting complex physical phenomena, with scenarios tailored for both Newtonian and non-Newtonian fluid behaviors, as well as thermodynamics in nuclear fusion.

Through experiments, the authors demonstrate that PhysPDE not only accurately recovers existing PDE terms but also extrapolates effectively to new scenarios, outperforming baseline models in terms of recall, precision, and computational efficiency.

**Strengths:**

The paper does make an original contribution to the state of knowledge, as it reconceptualizes PDE discovery as a physically guided interpretation task. This work focuses on **PDE interpretation**, which is the term they use to denote the integration of prior physical knowledge into the discovery process. By framing the task as one of **hypothesis selection**—with a goal to align with established physical laws rather than simply fit data—the authors address the crucial challenge of **physical interpretability** in machine learning for scientific discovery.

The paper is well written, with explanations offered with adequate rigor, clearly articulating the mixed-integer programming formulation and providing a detailed breakdown of the two-layer optimization process for hypothesis selection and expression recovery.

The offered **new datasets**are certainly of significance to the PDE, and particularly fluid mechanics and laser fusion, communities. The experimentation protocol is robust, involving cross-validation, varying boundary conditions, and multiple baselines for comparative analysis.

**Weaknesses:**

While the paper introduces a promising framework for integrating physical interpretability into PDE discovery, there are areas where the work could be improved to enhance its impact, applicability, and rigor. More specifically,

-  The paper offers a **comparative analysis** with traditional symbolic regression methods, such as PDE-FIND and GPT-based pipelines, however, it lacks an explicit comparison with recent interpretability-focused PDE discovery approaches, as for instance, the work by Champion et al. (2019), or Chen, Liu, Sun et al (2021), who adopt physics biases to discover interpretable models through sparse data.

- While PhysPDE introduces a bi-level optimization approach, there is limited discussion on how this specific approach advances the interpretability or robustness over similar frameworks like the sparse identification of nonlinear dynamical systems (SINDy), where laws are also indirectly accounted for in the selection of the basis functions.

- The paper provides an experimental evaluation of PhysPDE on relatively controlled (and wely generated) datasets, but the scalability of the framework for **high-dimensional, real-world systems** remains unclear. In practical applications such as climate modeling or biomedical engineering, PDE systems can have many more variables and parameters, leading to extremely large hypothesis spaces. The **Monte Carlo Tree Search (MCTS) and Sequential Threshold Optimization (STOP)** methods presented here may struggle with such scalability without further refinement.

- Although the authors mention that PhysPDE is tested under data noise, the experiments are still based on **synthetically generated datasets**, where the noise distribution can be controlled (as is here the case). Real-world data is typically noisier and may have **systematic errors, non-Gaussian noise, or missing data points**, which could impact the framework’s robustness. In this setting, reliance on a structured hypothesis decision forest could lead to inaccuracies, as the framework might overly rely on a narrow set of physical assumptions that may not fully capture complex, real-world phenomena.

- The **design of the hypothesis decision forest** is critical to PhysPDE’s interpretative power, yet the criteria for constructing this decision structure, such as the choice of hypotheses and decision hierarchy, are not discussed in detail. It is unclear how flexible or adaptable this forest structure is, which could limit the framework's usability across different scientific domains. For instance, if a user wanted to apply PhysPDE to a new domain like biomechanics, would they need to manually create a decision forest from scratch?

**Questions:**

Following up on the precious remarks, the following questions arise:

   -  How does PhysPDE compare specifically to recent frameworks that incorporate physical interpretability into PDE discovery, such as those seeking to discovery interpretable models from sparse data Champion et al. (2019), or Chen, Liu, Sun et al (2021)?
- How does PhysPDE handle the computational demands of hypothesis selection when scaling to high-dimensional systems, where the hypothesis space grows exponentially? Could the current approach handle systems with higher-dimensional PDEs or more complex, multi-parameter hypothesis spaces, such as those in climate or geophysical modeling?
- How robust is PhysPDE to real-world data noise, which may not follow Gaussian distributions or may include systematic measurement errors? Would the authors consider augmenting their experimental setup with various types of noise, or could they propose methods to make PhysPDE more robust to noise common in observational datasets?
-  Could you elaborate on how the hypothesis decision forest is constructed, specifically the criteria for selecting hypotheses and organizing them hierarchically? Consider sharing any automation or semi-supervised methods used in building this structure, as this could significantly impact PhysPDE’s ease of adaptation for other researchers.
- How does the decision-making process in the hypothesis selection step work, especially in cases where multiple hypotheses may fit the data similarly well? Also, if multiple plausible hypotheses exist, does PhysPDE allow for any probabilistic ranking or ensemble approach to indicate uncertainty, or is it a hard-selection process?
- Could the authors clarify how they evaluate interpretability and physical alignment of the recovered PDEs, beyond standard metrics like recall, precision, and MSR?

---

> ### Author Response · Authors · 2024-11-23
>
> We sincerely thank the reviewer for detailed and thoughtful feedback on our work. We are pleased that the reviewer acknowledges our contributions, including the introduction of the new PDE interpretation task and the bi-level optimization algorithm. Below, we carefully address the reviewer's comments point by point.
>
> ---
> **Q1. Comparison with interpretability-boosted PDE discovery**
>
> **A1.** Existing PDE discovery algorithms, only extract a symbolic expression from data. Certain frameworks, including Champion et al. (2019) and Chen et al. (2021), incorporate "physics" (more precisely, math) priors to boost the accuracy of expression. However, our dataset enables the algorithm to interpret the observation by physics hypothesis. For example, compressibility is an essential property of fluid (see textbook[1]), and one might expect the algorithm to tell whether the data is from compressible or incompressible fluid, instead of finding expressions under a given compressible fluid prior. Hence our method is more flexible and useful to real-world scientific research.
>
> ---
> **Q2. How to handle high-dimensional cases?**
>
> **A2.** High-dimensional problems primarily result in higher computational overhead. Our main contribution lies in the new task and the new dataset. While PhysPDE serves as a baseline model, it has not been specifically optimized for computational time (we provide some possible speedup techniques below). It can handle the presented datasets (search spaces of size 3000+ and 2D PDEs) within approximately 4 CPU hours which we think is affordable for many practical use.
>
> For higher-dimensional problems, it calls for modifications to the corresponding modules within the PhysPDE framework. For instance, if it is tailored for solving high-dimensional PDEs, the current finite difference algorithm could be replaced with alternatives such as PINN or DeepONet. Similarly, if solving combinations of more hypotheses (i.e., large-scale integer programming problems), deep learning-based methods (e.g. reference [2]) would be a promising alternative. We have to point out that solving the high-dimensional PDEs and large-scale integer programming are long-standing problems in the AI and math community, which is slightly beyond the scope of this benchmark paper.
>
>
> ---
> **Q3. How robust is PhysPDE to non-Gaussian noise**
>
> **A3.** We tested on multiple non-Gaussian noises to the S1 dataset and the performance of the top-1 decisions are shown in the table below. One can observe that PhysPDE is generally robust to various noises.
>
> > Table A3. PhysPDE S1 performance under various types of noise.
> >
> | Noise            | Precision | Recall | Test MSR              |
> | ---------------- | --------- | ------ | --------------------- |
> | -                | 1.00      | 1.00   | $4.83 \times 10^{-4}$ |
> | Pareto(a=1)      | 1.00      | 1.00   | $4.86 \times 10^{-4}$ |
> | Laplace(loc=0)   | 1.00      | 1.00   | $4.93 \times 10^{-4}$ |
> | Standard_t(dt=3) | 1.00      | 1.00   | $9.91 \times 10^{-4}$ |
>
> ---
> **Q4. How the hypothesis decision forest is constructed? Any automation or semi-supervised methods?**
>
> **A4.** The current method for hypothesis design is manual, extracting hypotheses from the table of contents of textbooks and the selection boxes of multi-physics solvers (e.g., COMSOL). We have attempted to use LLMs to automatically design hypotheses, but we found that the results contained logical and mathematical errors. One direction for future work might be to apply knowledge graph extraction to automatically extract hypotheses from textbooks and papers.
>
> ---
> **Q5. What if multiple plausible hypotheses exist? Probabilistic ranking or ensemble approach?**
>
>
> **A5.**  There are two mechanisms to handle the case of multiple plausible hypotheses:
> 1) Occam's Razor: simpler hypotheses are prioritized. This is implemented as a soft selection by adding complexity penalty terms (Eq. 3).
> 2) Ensemble Learning: The data is randomly divided into groups, and for each group, only the top 3 hypothesis tree search results are retained. The final results are selected based on the frequency of occurrence of the outcomes. The results reported in the paper are the ones with the highest frequency.
>
> ---
> **Q6. How to evaluate interpretability and physical alignment beyond standard metrics?**
>
> **A6.** Standard evaluation metrics find matching between mathematical expressions, with precision and recall defined for the set of terms in the expression. In comparison, we consider the matching degree of hypothesis selection, and the precision and recall in our paper are defined for the set of hypotheses. Based on our tree structure, one can also extend the evaluation metrics, such as using the shortest path length between decision nodes or the height of the lowest common ancestor as additional metrics.
>
> Reference:
>
> [1] White, Frank M., and Henry Xue. Fluid mechanics
>
> [2] Nair, Vinod, et al. Solving mixed integer programs using neural networks.

---

> > ### Comment · Reviewer_QbPL · 2024-11-24
> > **Feedback on revisions.**
> >
> > Thank you for your thoughtful responses and revisions. I appreciate the effort you put into addressing my comments. After considering your updates, I believe my initial assessment still stands, as the changes do not significantly impact the concerns I raises. Existing schemes may indeed try to see in physics explicitly or more loosely by imposing constraints like thermodynamics, sympleciticity, ect. It is not clear how this work is indeed something conceptually different. However, my overall appreciation for this work, allows me to maintain my rather high grades as is.

---

> > > ### Author Response · Authors · 2024-11-25
> > >
> > > Dear reviewer QbPL:
> > >
> > > Thank you for your valuable feedback. We want to clarify that the key difference is the task definition.
> > >
> > > ---
> > > - The existing symbolic regression schemes, although with physics constraints, aim to regress **only** the math expression. The physics priors are manually added, and can not change during the search. The example outputs are:
> > >     +  $u_t=-uu_x + 0.1u_{xx}$,  in Chen et al. (2021) Fig 2.,
> > >     + $z_{tt} = -\sin z$,  in Champion et al. (2019) Fig 2.,
> > >
> > >     without any physics interpretation produced by the algorithm.
> > >
> > > ---
> > > - Our physics interpretation task is to search **both** math expression and interpretation.  In the above example of Chen et al. (2021), given fluid mechanics priors, our algorithm output might be:
> > >     + $u_t=-uu_x + 0.1u_{xx},  \quad H=[\text{viscosity=True,  body force=False}].$,
> > >
> > >   where the $H$ is the hypothesis the algorithm searched to interpret the data. The viscosity corresponds to the $u_{xx}$, and the body force is an external force $F$ not shown here.
> > > One can observe that the existing schemes can not be directly applied to our task, even with some pre-defined physics constraints.
> > >
> > > Thanks again for your time. We hope that our responses satisfy your requirements, and we are looking forward to your further feedback.
> > >
> > >
> > > Best regards,
> > >
> > > PhysPDE Authors

---

> > > > ### Comment · Reviewer_QbPL · 2024-11-29
> > > >
> > > > Thank you for the clarification. This is already clear and has been taken into account in my assessment. My point is that the choice of activating physics-based biases (or constraints) can easily be added to those further schemes (e.g. in the form of a model selection problem)- This can then  regulate the way in which appropriate dictionaries, or bases are selected.
> > > > I do still find value in your suggestion, hence the high rating despite the expressed concerns.

---

> > > > > ### Author Response · Authors · 2024-11-30
> > > > >
> > > > > Dear Reviewer QbPL,
> > > > >
> > > > > Thank you for your thoughtful comments and the positive feedback despite the concerns raised. We appreciate your point that the existing physics-based PDE discovery algorithms can incorporate physics model selection. This might suggest that existing PDE discovery algorithms can be easily applied to our benchmark, and that our HSTS+STOP baseline is an alternative in this context.
> > > > >
> > > > > However, we would like to emphasize that our key contribution is the PhysPDE task and benchmark. The point that existing algorithms can be easily adapted to the benchmark shows the versatility and potential of the PhysPDE. We believe the PhysPDE benchmark will provide a foundation for assessing and improving many ML4Sci algorithms, not limited to the HSTS+STOP baseline.
> > > > >
> > > > > We hope this clarification strengthens the contribution of our work and appreciate your recognition of the value of our suggestion.
> > > > >
> > > > > Best regards,
> > > > >
> > > > > PhysPDE Authors

---

> > > > > > ### Comment · Reviewer_A96C · 2024-11-30
> > > > > >
> > > > > > Hello Reviewer QbPL, I want to weigh in here. Hello authors, despite my good score of 8, I am not quite sure how this paper can be a benchmark for other algorithms. I think this paper is more like a new method that incorporates physics in a way that serves well the purpose of aligning with existing physical principles. I do not agree with the phrase "other algorithms can be applied to the benchmark". Benchmark should be more like the comprehensive evaluation of existing methods, rather than a new method that is essentially different from existing works, especially those vanilla symbolic regression and tree-based search models. If you want to claim novelty as benchmark, you will need to do a much more comprehensive experimental evaluation for all the important published works in this venue, or demonstrate, not just saying, how other existing algorithms can adapt your work.

---

> > > > > > > ### Author Response · Authors · 2024-12-01
> > > > > > > **Response to Reviewer A96C**
> > > > > > >
> > > > > > > Thank you for your thoughtful feedback and for taking the time to carefully review our manuscript and rebuttal. We will clarify two points in response to your feedback:
> > > > > > >
> > > > > > >
> > > > > > > **Q1. What is a 'Benchmark'? Is the benchmark our main novelty?**
> > > > > > >
> > > > > > > **A1:** We understand Reviewer A96C's point that a benchmark should be a comprehensive evaluation of existing methods. We acknowledge that PhysPDE may not fully align with the traditional definition of a benchmark paper. However, in a broader sense, we believe that a benchmark typically includes the following components:
> > > > > > > 1) a dataset (e.g., Fluid Mechanics dataset);
> > > > > > > 2) a task setting (e.g., Physics Interpretation task);
> > > > > > > 3) a workflow for evaluation (e.g., integrated/pipelined approaches as shown in Figure 1);
> > > > > > > 4) evaluation results.
> > > > > > >
> > > > > > > In our paper, our primary contributions lie in components 1, 2, and (the new algorithm implemented for) 3. Thus, we acknowledge that our previous statement, "our contribution is the task and benchmark," was somewhat imprecise. A more accurate description, as stated on page 2 of the draft, would be: "our contributions are the dataset, task, and an algorithm."
> > > > > > >
> > > > > > > ---
> > > > > > >
> > > > > > > **Q2. Is the phrase "other algorithms can be applied to the benchmark" true?**
> > > > > > >
> > > > > > > **A2:** This phrase is indeed a central concern raised by Reviewer QbPL. While we have discussed and provided examples showing that existing algorithms cannot be directly applied to our approach, QbPL further suggests that with some modifications, existing algorithms could be adapted. We partially agree with this observation, as our baseline method, SINDy+GPT, is itself a modification of existing methods, albeit a non-trivial one. Therefore, in our previous rebuttal, we phrased it as "This **might** suggest that existing algorithms can be more easily applied." In this context, we have reached a consensus with QbPL that our work still offers significant value as a benchmark (in the broader sense).
> > > > > > >
> > > > > > > ---
> > > > > > > We have appended the discussion above to the paper draft and temporarily uploaded it to an anonymous platform ([link](https://anonymous.4open.science/r/ICLR-2025-PhysPDE-Draft-CD30/ICLR_2025___PhysPDE.pdf)). Once again, thank you for your insightful feedback and for helping us refine our work. We hope the clarifications provided above address your concerns.

---

### Official Review · Reviewer_QC1C · 2024-11-06

**Soundness:** 2
**Presentation:** 2
**Contribution:** 2
**Rating:** 6
**Confidence:** 4

**Summary:**

The authors introduce a new task of explaining observed data using existing partial differential equations in physics and other domains. Previous approaches to discover new partial differential equations via symbolic regressions and randomized algorithms starts from scratch. However, PDEs are inspired from domain knowledge and available theories. The authors propose a method HSTS and STOP algorithm that can find/construct appropriate and interpretable PDEs, incorporating available domain knowledge. Experiments show that the authors proposed method provide better performance compared to baseline methods.

**Strengths:**

The motivation of the work is described well in the introduction.

**Weaknesses:**

The method proposed by the authors needs to be explained in more detail, using an example if possible. Figure 2 caption needs to include more details, for example difference between learned PDE and discovered new expression is not clear. Figure 3 caption needs more details, are these learned from the author's algorithm or available from domain knowledge?

**Questions:**

Although the author's motivation is to generate interpretable PDE models on top of existing domain knowledge, is this work limited to discovering only known theories? For example, given the knowledge of Newtonian fluid mechanics, can this method discover PDEs for non-newtonian fluids?
Another thing to mention here is there are previous approaches for empirical law discovery such as BACON. What benefit does the author's method provide compared to such previous works?
Additionally, how do the authors see the dataset being used in future works? Given these datasets, is there a metric that can measure the interpretability of PDEs discovered by different algorithms, or do we still need domain expert opinion on that?

---

> ### Author Response · Authors · 2024-11-23
>
> We sincerely thank the reviewer for the thoughtful feedback and constructive comments. We are delighted that the reviewer recognizes the novelty of our work in addressing the new task of explaining observed data using physics domain knowledge and theories in this context. Below, we address the specific questions and suggestions raised by the reviewer in detail.
>
> ---
> **W1. need more explanation in method, figure 2 and figure 3.**
>
> **A0.** We have modified the draft based on your suggestions. The modification in the main text is marked in blue in the updated draft ([link](https://openreview.net/pdf?id=G3CpBCQwNh)). A detailed explanation of the algorithm is in Appendix E&F. We also updated a new version of Figure 3 ([link](https://postimg.cc/230gnr81)), with an example of constructing a PDE system from hypotheses.
>
> ---
> **Q1. is this work limited to discovering only known theories**
>
> **A1.** Our work can discover new expressions on top of the known theories, if new expressions (empirical formulas) could be regarded as a kind of new theory. The example you mentioned, discovering unknown non-Newtonian fluids given knowledge of Newtonian fluid, is exactly the S2 experiment in the paper (see the second column of Table 9).
>
>
> **Q2.  benefits compared to previous empirical law discovery algorithms**
>
> **A2.** The benefits are at least twofold: interpretability and empirical performance.
>
> 1) previous empirical law discovery algorithms like BACON only produce pure math expressions like y=f(x), without any physical interpretation such as Newton's laws of motion, Fourier's law of heat conduction, etc. Our work is based on the physical hypothesis search and produces a physical interpretation of data, which is critical for real-world physics research.
>
> 2) the incorporation of physics laws boosts the model's performance. As shown in Table 5-8, our model outperforms existing math-only models with a clear margin in both accuracy and simplicity.
>
> ---
> **Q3. future use of the dataset**
>
> **A3.** The dataset can be used in both AI and physics communities.
>
> - **AI.**
> The newly defined physical hypothesis discovery task can encourage the design of models with more physical interpretability. The two datasets we provide serve as a benchmark for these new models. For instance, our current experiments demonstrate that GPT-4o (Table 10) performs poorly on the physical hypothesis selection task. A potential research direction is to explore how to improve existing large language models to handle such tasks and datasets, thereby advancing the development of AI-powered physics research assistants.
>
> - **Physics.**
> In the field of physics, experts can leverage their domain knowledge alongside our existing frameworks and algorithms to conduct cutting-edge scientific exploration. For example, by incorporating radiation heat transfer theory into fluid dynamics, researchers can investigate the mechanisms of nuclear fusion reactions (Section 5). The model is capable of automatically selecting the most plausible physical hypotheses based on experimental or simulation data, offering both inspiration and evidence for scientific discovery.
>
> ---
> **Q4. is there a metric that can measure the interpretability, or do we need domain expert opinion?**
>
> **A4.** We do not rely on domain expert opinion on the measurement. The evaluation metrics related to interpretability in this benchmark are objective and quantitative:
> 1) Match of hypothesis with the truth, evaluated by precision and recall between the selected and true hypothesis. In the future, one can also extend the evaluation metrics based on our tree structure, such as the shortest path length between decision nodes or the height of the lowest common ancestor.
>
> 2) Complexity of hypothesis, evaluated by the size of the hypothesis set.  The hypothesis should be as simple as possible, according to the Occam's razor.

---

### Meta-Review · Area_Chair_uGDn · 2024-12-28

**Metareview:**

This work presents a new set of benchmarks and novel tasks that are highly relevant to the promising field of machine learning for science.  Novel datasets with experimental data for fluid mechanics and laser fusion. While the primary motivation is to derive mathematical (PDE-based) expressions from observational data alongside physical laws and hypotheses, rather than simply identifying expressions that model the data well (e.g., via symbolic regression).  As highlighted by the authors benchmarks, and research that makes progress against these benchmarks, benefit both the ML communities (leveraging physical knowledge in PDE discovery, which can improve search efficiency and interpretability) and physics communities (who can leverage these tools, especially as they advance).  The authors propose a number of intriguing directions for their work, including incorporation of LLMs and applications to important areas of study such as nuclear fusion.  Reviewers found the idea of having physically-grounded and interpretable discovery of PDEs to be compelling, and the most detailed reviews also commented on the thoroughness of the dataset construction.  Having read this paper in some detail, I agree with these sentiments, and advocate for acceptance.

**Additional Comments On Reviewer Discussion:**

There were access issues to the dataset that were promptly resolved after I raised the issue with the authors.  The repository appears to be complete and satisfies reviewers, including Xb5C who initially had difficulty accessing the data.  Overall reviewers were positive on the paper, all advocating for acceptance.  Most of the weaknesses stated by the reviewers were clarification questions that the authors answered.  The authors also provided a clear summary of their contributions and responses to the author critiques.

---

### Decision · Program_Chairs · 2025-01-22

Accept (Poster)